

# Understanding the Relative Importance of Vertical and Horizontal Flow in Ice-Wedge Polygons

Nathan A. Wales[1,2], Jesus D. Gomez-Velez[2,3], Brent D. Newman[1], Cathy J. Wilson[1], Baptiste Dafflon[4], Timothy J. Kneafsey[4], Stan D. Wullschleger[5]

[1]Los Alamos National Laboratory, Los Alamos, NM, 87545, USA
[2]Hydrology Program, Department of Earth & Environmental Science, New Mexico Institute of Mining and Technology, Socorro, NM, 87801, USA
[3]Department of Civil and Environmental Engineering, Vanderbilt University, Nashville, TN, 37235, USA
[4]Lawrence Berkeley National Laboratory, Berkeley, CA, 94720, USA
[5]Oak Ridge National Laboratory, Oak Ridge, TN, 37831-6301, USA

*Correspondence to*: Nathan A. Wales (nathanwales@gmail.com)

**Abstract.** Ice-wedge polygons are common Arctic landforms. The future of these landforms in a warming climate depends on the bidirectional feedback between the rate of ice-wedge degradation and changes in hydrological characteristics. This work aims to better understand the relative roles of vertical and horizontal water fluxes in the subsurface of polygonal landscapes, providing

new insights and data to test and calibrate hydrology models. Field-scale investigations were conducted at an intensively-instrumented location on the Barrow Environmental Observatory (BEO) near Utqiaġvik, AK, USA. Using a conservative tracer, we examined controls of microtopography and the frost table on subsurface flow and transport within a low-centered and a high-centered polygon. Bromide tracer was applied at both polygons in July 2015 and transport was monitored through two thaw seasons. Samplers arrays placed in polygon centers, rims, and troughs were used to monitor tracer concentrations. In both

polygons, the tracer first infiltrated vertically until encountering the frost table, then was transported horizontally. Horizontal flow occurred in more locations and at higher velocities of fluxes in the low-centered polygon than in the high-centered polygon. Preferential flow, influenced by frost table topography, was significant between polygon centers and troughs. Estimates of horizontal hydraulic conductivity were within the range of previous estimates of vertical conductivity, highlighting the importance of horizontal flow in these systems. This work forms a basis for understanding complexity of flow in polygonal landscapes.

## 1 Introduction

A mechanistic understanding of the feedbacks between Arctic climate and terrestrial ecosystems is critical to understand and predict future changes in these sensitive ecosystems. Observations suggest that high latitude systems are experiencing the most rapid rates of warming on Earth, leading to increased permafrost temperatures, melting of ground ice, and accelerated permafrost degradation (Hinzman et al., 2013; Jorgenson et al., 2010; Romanovsky et al., 2010). Permafrost degradation is of primary concern

in the Arctic, as it affects hydrology, biogeochemical transformations, and human infrastructure (Andersland et al., 2003; Heikoop et al., 2015; Hinzman et al., 2013; Jorgenson et al., 2010; Lara et al., 2015; Liljedahl et al., 2011; Newman et al., 2015; Zona et al., 2011a). The northern Arctic permafrost zone covers twenty-four percent of the landmass in the northern hemisphere and stores an estimated 1.7 billion tons of organic carbon (Hugelius et al., 2013; Schuur et al., 2008, 2015; Tarnocai et al., 2009; Zimov et al., 2006) with a significant fraction stored in the Arctic tundra, where ice-wedge polygons are among the most prolific

geomorphological features (Hussey and Michelson, 1966). Degree of soil saturation influences whether carbon is released as carbon dioxide or methane. Thus highlighting the importance of understanding the hydrology of permafrost regions.

Ice-wedge polygons form as thermal contraction creates cracks in the ground. Each year, with spring snowmelt, these cracks collect water, which subsequently freezes to form an ice-wedge below the surface (Liljedahl et al., 2016). Over time, the ice-wedge grows, displacing ground and eventually forming a low-centered polygon (Fig. 1). When ice-wedges around a low-


centered polygon degrade, the ground above them subsides, inverting the topography and creating a high-centered polygon (Gamon et al., 2012; Jorgenson and Osterkamp, 2005). These two polygon types represent the geomorphological end members of ice-wedge polygons. All polygons have three primary microtopographic features: centers, rims, and throughs. A low-centered polygon is defined as an ice-wedge polygon with the topographic low at the center and is characterized by low, saturated centers and troughs

with high and relatively dry rims. A high-centered polygon is defined as an ice-wedge polygon with the topographic high at the center and is characterized by low, saturated troughs and high, dry centers and rims.

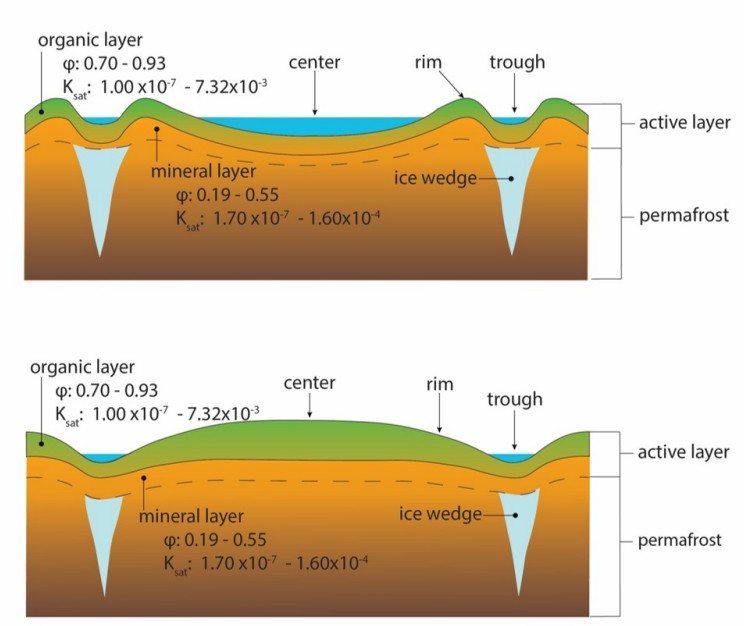

**Figure 1. Conceptual diagram of a low-centered polygon (top) and a high-centered polygon (bottom). Porosity and Ksat [m·s-1] values from literature (Atchley et al., 2015, 2015; Beringer et al., 2001; Hinzman et al., 1991, 1998; Lawrence and Slater, 2008; Nicolsky et al.,**
**2009; O'Donnell et al., 2009; Price et al., 2008; Quinton et al., 2000).**

      It is now established that significant hydrological and biogeochemical differences exist on the sub-meter scale and are influenced by ice-wedge polygon type and microtopographic feature (Andresen et al., 2016; Lara et al., 2015; Liljedahl et al., 2016; Newman et al., 2015; Wainwright et al., 2015). Permafrost degradation has the potential not only to change microtopographic

features of ice-wedge polygons, but their hydrologic regimes as well. Understanding hydrologic regimes can help determine the fate of organic matter and nutrients in these landforms. For example, whether organic matter is decomposed into carbon dioxide or methane is largely determined by local hydrology.

      While many studies have focused specifically on ice-wedge polygons, (Boike et al., 2008; Heikoop et al., 2015; Jorgenson et al., 2010; Lara et al., 2015; Newman et al., 2015), no studies, to our knowledge, have been conducted toward quantifying the

relative roles of subsurface vertical and horizontal fluxes, or characterizing heterogeneity of subsurface flow and transport within these landforms. Furthermore, most regional and pan-Arctic land models ignore horizontal flux and focus only on the representation of vertical water fluxes in the form of infiltration and evapotranspiration (Chadburn et al., 2015a, 2015b; Clark et al., 2015). There exists a need to better quantify the relative roles of vertical and horizontal subsurface water fluxes in these landscapes, providing insight and data to revise, test, and calibrate permafrost hydrology representations in these models.

To this end, a tracer study was conducted on polygonal ground in the Barrow Peninsula of Alaska from July 2015 to September 2016. The Barrow Peninsula is located on the Arctic Coastal Plain adjacent to the Arctic Ocean. Approximately 65%



of the land cover in the Barrow Peninsula is ice-wedge polygonal ground, making this an ideal place to study the hydrology of ice-wedge polygons (Bockheim and Hinkel, 2010). To the best of our knowledge, and with the exception of an invasive and localized dye tracer experiment (Boike et al., 2008), this is the first non-invasive tracer study to be conducted at the polygon scale. Furthermore, this experiment is unique in that a tracer was continually monitored simultaneously on both low- and high-centered

polygons throughout thaw seasons, making it possible to characterize the breakthrough curves and determine times of first arrivals. Therefore, our approach permits a comparison of behaviors in low- and high-centered polygons over the same time period and meteorological conditions.

The purpose of this paper is to examine how differently low- and high-centered polygons behave hydrologically, and evaluate the relative importance of vertical and horizontal flux within polygon systems (including the controls of the frost table

and microtopography on subsurface hydrology). The presence of significant horizontal flow can guide new upscaling approaches to incorporate these landscape features into regional hydrologic and biogeochemical models, which traditionally conceptualize the subsurface flow within ice-wedge polygons as exclusively vertical (Chadburn et al., 2015b; Clark et al., 2015). Insights from this study are intended to inform future work on the possible effects of permafrost degradation by improving the conceptualization used in the Arctic Terrestrial Simulator, developed by the Department of Energy at Los Alamos National Laboratory (Atchley et

al., 2015; Painter et al., 2016). Our primary focus is the hydrology of the active layer, which is the portion of the soil profile that thaws each year (Hinzman et al., 1991), with some emphasis on surface water. Possible mechanisms of flow heterogeneity are also discussed.

## 2 Materials and Methods

### 2.1 Site Description

The study site is located east of Utqiaġvik (formerly Barrow), AK, USA on the Arctic Coastal Plain in the Barrow Environmental Observatory (Fig. 2). Climate of this region is characterized by long winters, short summers, with a mean average annual temperature of -10.2 °C, and mean annual precipitation of 141.5 mm (NOAA-NCDC, 2000-2016). Coldest temperatures occur in February with warmest temperatures in July (NOAA-NCDC, 2000-2016). The thaw season usually begins in June with maximum thaw depth occurring sometime in late August or early September. Freeze up typically begins sometime in September,

subsequently leaving the ground completely frozen until June when the next thaw season begins. After the brief snowmelt period, a receding water table despite precipitation indicates that evapotranspiration dominates during the first half of the thaw season while a rising water table with precipitation indicates that precipitation and infiltration dominate during the second half of the thaw season. These observations are consistent with observations of evapotranspiration during the two years prior to the tracer experiment described here (Raz-Yaseef et al., 2017) and in other previous studies on Arctic water balances (Helbig et al., 2013;

Pohl et al., 2009).

The region is characterized by low relief land forms underlain by continuous, perennially frozen permafrost >400 m thick and an active layer depth ranging from 30-90 cm (Hinkel et al., 2003; Hubbard et al., 2013). The soil profile consists of an organic layer typically <40 cm thick underlain by a silty mineral layer composed primarily of quartz and chert (Black, 1964; Hinkel et al., 2003). Volume of shallow ground ice in the region can be as high as 80% and is comprised primarily of ice-wedges and cryogenic

structures (Kanevskiy et al., 2013). Patterns of cryogenic structures found in frozen soils can result in higher porosities than found in unfrozen soils (Dafflon et al., 2016).



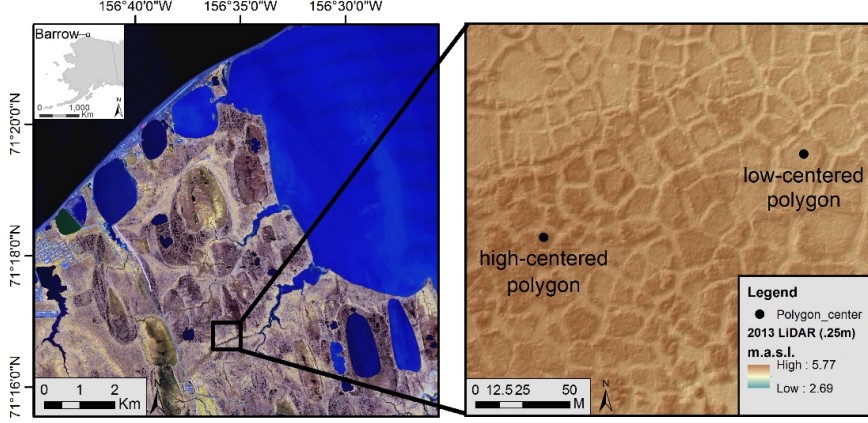

**Figure 2. Map showing the portion of the Barrow Peninsula where the study was conducted (left) and a close up of the area containing the low- and high-centered polygons used for the tracer study.**

5    One high-center polygon (with an area of 132 m$^2$) and one low-center polygon (with an area of 706 m$^2$) were chosen to reflect the extremes of tundra polygon morphology (Fig. 3). Only two polygons were used to minimize anthropogenic perturbations to the study site and because the cost and logistical complexity of these experiments is significant. Even though this limits our ability to replicate the results, the polygons selected are representative of a larger inventory of low- and high-center polygons being investigated by our team at this intensive study site, providing new insight into hydrologic differences between polygon types and

10   into flow and transport across polygon features. The general soil profile of the polygons was an organic layer, comprised of 2-20 cm moss and peat (Iversen et al., 2015), underlain by a seasonally thawed mineral soil layer, followed by permafrost (Fig. 1).

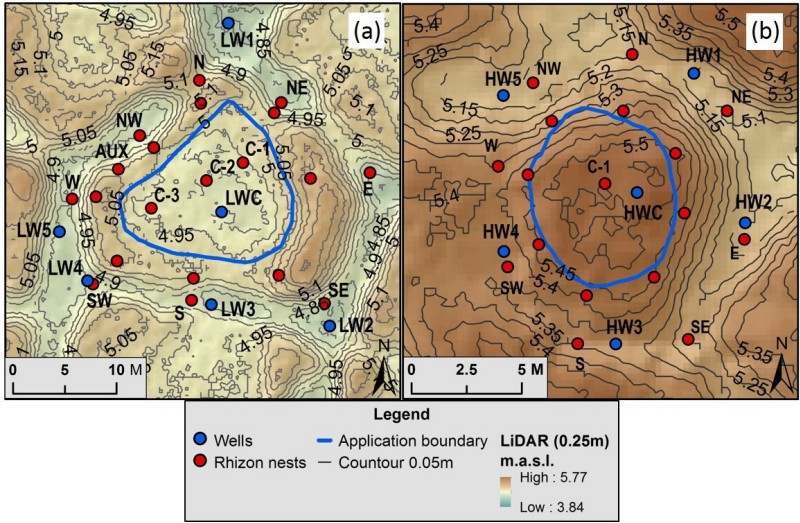

**Figure 3. Digital elevation model for the low-centered (a) and high-centered (b) polygons. Red dots represent locations of sampler nests**
15   **and blue dots represent locations of observation wells. Blue circle indicates area of tracer application and encompasses the polygon center. Note scales are different for the two polygons.**


**2.2 Observational Network**

Each polygon had been instrumented with six fully-screened observation wells (3.81 cm diameter PVC casing), one in the center of the polygon and five distributed along the surrounding troughs (blue circles in Fig. 3). Pressure transducers (Diver, Schlumberger Water Services, Netherlands) were deployed in each well to measure stage fluctuations at fifteen-minute intervals and used to

estimate water table elevations relative to ground surface (measured in meters above sea level, masl). The pressure transducers have an accuracy of ±0.5 cm-H₂O and a resolution of 0.2 cm-H₂O.

To prevent preferential flow along the well casings, 15.24 cm diameter PVC pipe was placed around each well casing and pressed through the organic layer into the top 2 centimeters of the mineral layer (Fig. 4b). Silicon sheets 30.5 cm × 30.5 cm × 0.24 cm with pre-cut holes were also placed around each 15.24 cm pipe at the ground level to form a watertight seal. Additional

silicon sheets of the same dimensions were placed around the samplers (discussed below) to prevent preferential/wall flow along the outer casing of the samplers. Caps were also placed on samplers between sampling events to prevent precipitation from collecting inside the housing of the samplers and diluting samples.

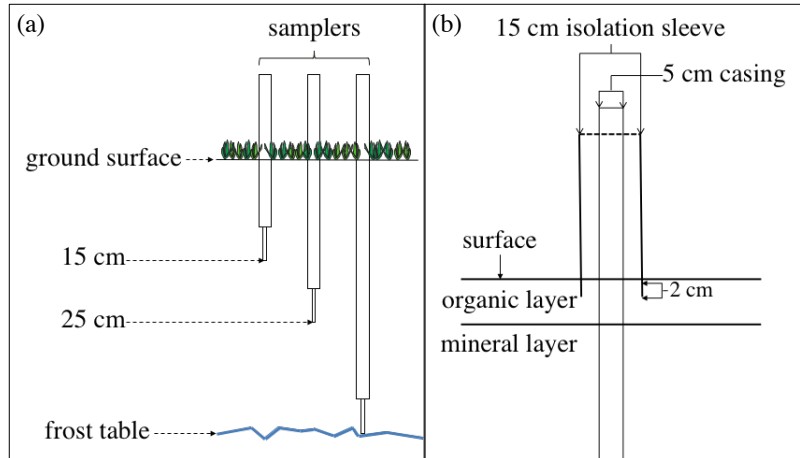

**Figure 4. (a) Schematic representation of a Rhizon sampling nest and (b) a schematic representation of the observation wells with isolation sleeve.**

MacroRhizon samplers (Rhizosphere Research Products, Netherlands) were used to sample pore water at various locations and depths in both study sites (Fig. 4a). These samplers minimize perturbations to the porous media matrix and flow field

by collecting sample volumes at low rates, no greater than 60 ml day⁻¹, driven by the suction of a syringe at the surface. In addition, the sampler dimensions made it feasible to simultaneously sample three soil depths within a 12 cm diameter circle. Each sampler collected water through a tip 9 cm in length and 4.5 mm diameter with a mean pore size of 0.15 μm. In this work, sample depths refer to the insertion depth of sampler tip ends (Fig. 4a). In most cases, syringes remained on the samplers overnight to collect sufficient sample volumes, and therefore some sampling periods spanned over 24 hours. When freezing temperatures were

expected overnight, sampling was initiated and collected on the same day.

The rims of each polygon had 8 nests of MacroRhizon samplers oriented in a radial pattern around the polygon (Fig. 3). Each sampler nest had samplers at 3 depths: 15 cm, 25 cm, and at the frost table. Samplers at the 15 cm and 25 cm depths were fixed over time while the deepest sampler, installed once the frost table reached a depth of 35 cm, was moved downward as the frost table depth increased. Troughs surrounding each polygon also had 8 nests of samplers adjacent to corresponding sampler

nests on the rims (Fig. 3). Three nests of samplers were placed in the center of the low-center polygon and only one in the center



of the high-center polygon due to its smaller relative area. Unlike the sampling nests in rims and troughs, samplers in polygon centers were inserted at 45 degrees so the sampling tips would protrude past the edges of the silicon sheets. Samplers in polygon centers were inserted to depths of 15 cm, 25 cm, and frost table depth was sampled at 35 cm and deeper (Fig. 4a). An additional sampler nest was placed in the rim of the low-center polygon where a saddle occurred, constituting an area of interest due to

possible flow convergence. To minimize perturbations and avoid the generation of preferential flow paths, samplers at frost table depth were not removed prior to freeze-up at the end of the 2015 thaw season. As a result, in 2016, the deepest samplers were not sampled until the frost table reached the deepest depth for 2015.

### 2.3 Bromide Tracer Test

Bromide was used a tracer due to its conservative nature with low potential for adsorption and ion exchange and negligible
background concentrations (Davis et al., 1980). Other tracers were considered, but low background levels of bromide had been previously established (Newman et al., 2015), and given the high organic matter content and low pH of active layer waters, bromide was thought to be the best option. Potassium bromide (KBr) was dissolved in water and applied to the center of each polygon with a garden sprayer (area of tracer application is outlined in blue Fig. 3). A reference grid of nylon cord was used to guide the even distribution of tracer, and disposable rubber booties and latex gloves were worn during application to prevent contamination outside
of the application area. Eight liters of tracer solution with a concentration of 5,000 mg l$^{-1}$ (40 g of Br) were applied to the high-center polygon on 12 July 2015 and 24 L of tracer solution with a concentration of 10,000 mg l$^{-1}$ (240 g of Br) were applied to the low-center polygon on 13 July 2015. The higher concentration and volume used in the low-centered polygon compensate for the surface area, about three times larger than the high-centered polygon, and the dilution caused by ponded water. Ten liters of water were subsequently sprayed on each polygon to facilitate infiltration of tracer into the soil.

### 2.4 Sampling and Analytical Methods

Sampling frequency varied depending on precipitation events and observed tracer concentrations. In 2015, samples were typically taken every two and four days during periods with and without precipitation events, respectively. We sampled daily during periods of persistent daily precipitation events. A full suite of samples was taken prior to tracer application to establish background levels of bromide. Pre-deployment bromide concentrations were consistent with those previously observed in the area (Newman et al.,
2015), and many pre-deployment concentrations were at or near the limit of detection of the ion chromatograph used for analysis (0.01 ppm). In addition to groundwater samples, grab samples of surface waters were also collected during each sampling event. Samples were frozen and shipped to the Geochemistry and Geomaterials Research Laboratory (GGRL) at Los Alamos National Laboratory (LANL) for analysis. Samples were thawed and filtered through a 0.45 µm syringe filter prior to analysis via ion chromatography with an uncertainty of ±5%.

Frost table depth measurements, taken with a tile probe, were typically taken weekly to the nearest 0.5 cm at each sampler nest. This served the dual purpose of ensuring the deepest sampler was at the depth of the frost table and measuring frost table depth. In both polygons, the frost table generally reached its deepest point in the beginning of September. Within the low-centered polygon, the maximum frost table depth measured over the two thaw seasons was 43 cm in the center, 45 cm in the rims, and 50 cm in the troughs. The maximum measured frost table depths for the high-centered polygon were 45 cm in the center, 43.5 cm in
the rims, and 38 cm in the troughs for both field seasons.

### 2.5 Ground Penetrating Radar

Ground penetrating radar (GPR) surveys were conducted on each polygon to understand the influence of frost table topography on flow. GPR has been used for various applications in Arctic regions including estimation of thaw layer thickness (Bradford et al.,





2005; Hubbard et al., 2013), characterization of permafrost and ice-wedges structure (Léger et al., 2017; Munroe et al., 2007) and mapping of snow thickness (Wainwright et al., 2017). In this study, common-offset surface GPR transects were collected on October 2, 2015 to estimate thaw layer thickness at the low- and high-center polygon locations. GPR data were collected using a Mala Ramac system with 500 MHz antennas along four ~34-m-long parallel transects crossing the low-centered polygon, and

along fifty-one ~15-m-long SE-NW transects spaced 0.25 m apart crossing the high-centered polygon. A wheel odometer was used to acquire traces with a spacing of 0.06 m. Minimal processing of the common offset lines included zero-time adjustment, bandpass filtering, automatic gain control, semi-automated picking of the two-way travel time to the key reflector, and conversion of travel time to depth. The key reflector corresponds to the interface between the thaw layer and the permafrost, as confirmed by the strong relationship between the GPR signal travel time and manual probe-based measurements of thaw layer thickness

(correlation coefficient ~ 0.73). The relationship has been used to convert the GPR signal travel time to thaw layer thickness. Frost table elevation was obtained by subtracting the GPR-inferred thaw layer thickness from the digital elevation model of the study site. Given the high spatial density of GPR data at the high-center polygon location, a frost table elevation map was obtained through linear interpolation.

### 2.6 Core Analyses

Shallow cores were extracted, using a SIPRE auger, from areas adjacent to the polygon tracer studies. Each core was collected from the frozen active layer at a different location. Cores were 46 mm in diameter and the lengths varied. Cores were kept frozen, and a few days after drilling, transported frozen to Lawrence Berkeley National Laboratory (LBNL) in Berkeley, CA. Three-dimensional images of the cores were obtained using a medical X-ray computed tomography (CT) scanner at 120 kV. Images were reconstructed to resolutions of 2.56 pixels per mm or better in the core-horizontal plane and 0.625 mm along the core-vertical axis.

Additional cores containing ice lenses were extracted from the frozen active layer using a 51 mm diameter AMS Soil Auger. Cores were kept frozen until subsampling at the Permafrost Laboratory at the University of Alaska Fairbanks.

### 2.7 Well Response and Recovery

To better understand the response of the polygons to precipitation inputs, we focused on the temporal characteristics of water level changes caused by 14 precipitation events occurring over the 2015 and 2016 thaw seasons (Fig. 5). Each of these events is preceded

and followed by relatively dry periods resulting in water level changes with clear ascending and recovery curves. Isolating the water level hydrograph associated to each precipitation event allowed us to estimate the maximum change in head ($\Delta h$), time-to-peak ($T_{peak}$), and characteristic recession time ($\lambda$) (Table 1). The characteristic recession time is calculated as the reciprocal of the slope for the line fitted to the natural log of water table elevation versus time during the recession limb. This recession time is a simple measure of the memory of the well to perturbations caused by precipitation events (Troch et al., 2013).






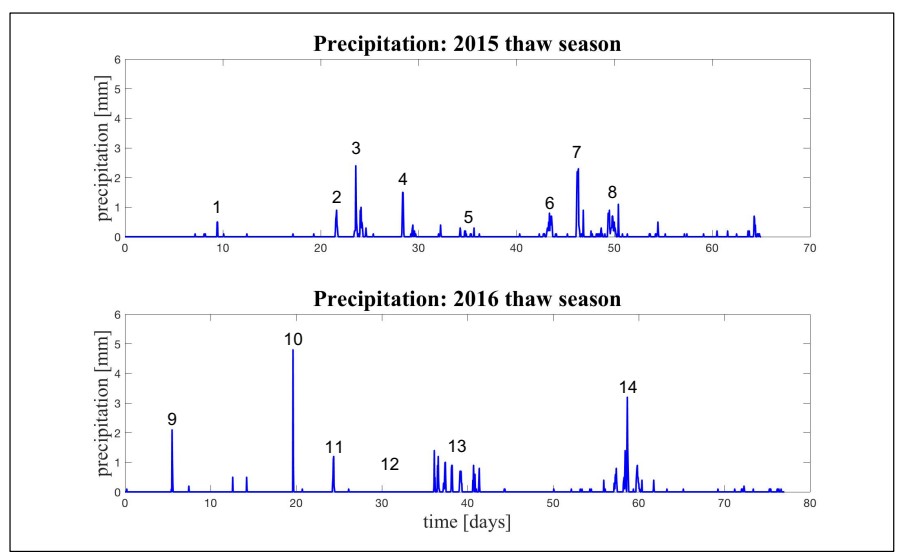

**Figure 5. Precipitation events used in the calculation of characteristics of well response. Note that event 12 is not a precipitation event, but marks where a recession limb was analyzed for each well.**

### 2.8 Tracer Arrival and Hydraulic Conductivity

5   The temporal evolution of tracer concentrations at selected observation wells was used to approximate average linear velocities and bulk hydraulic conductivity values for each polygon. To this end, velocities were estimated by assuming that the transport of the tracer within the polygons can be approximated as a one-dimensional advective-dispersive problem with adsorption effects – a reasonable assumption given the lack of information and uncertainty in the spatial distribution of hydraulic parameters. Van Genuchten and Alves (1982) found an analytical solution to this problem for the case of a semi-infinite soil profile without

10  production or decay and with a constant initial concentration:

$$c(x,t) = \begin{cases} C_i + (C_0 - C_i)A(x,t) & 0 < t < t_0 \\ C_i + (C_0 - C_i)A(x,t) - C_0 A(x, t - t_0) & t > t_0 \end{cases} \tag{1}$$

with

$$A(x,t) = 0.5 \, erfc\left[\frac{Rx - vt}{2(DRt)^{0.5}}\right] + 0.5 \, experfc\left[\frac{Rx + vt}{2(DRt)^{0.5}}\right] \tag{2}$$

and initial and boundary conditions given by

$$c(x,0) = C_i$$
$$c(0,t) = \begin{cases} C_0 & 0 < t < t_0 \\ 0 & t > t_0 \end{cases}$$
$$\frac{dc}{dt}(\infty,t) = 0$$





In Eqautions (1) and (2), $C_i$ [mg/l] is initial concentration, $C_o$ [mg/l] is input concentration, $t$ [hr] is time, $t_o$ [hr] is duration of solute pulse, $x$ [cm] is distance. The function $A(x,t)$ is the effluent concentration where $R$ is the retardation coefficient [-], $v$ is pore-water velocity [cm/hr], and $D$ [cm²/hr] is the dispersion coefficient.

We use the conditions in the field experiment to parameterize Eqs. (1) and (2). Background concentrations and tracer injections varied for each polygon (see Table 2). In the high-centered polygon, the background concentration was $C_i = 0.19$ mg/l and a solution with concentration $C_o = 5,000$ mg/l was injected over a period of $t_o = 4.5$ hours. On the other hand, the low-centered polygon had a background concentration of $C_i = 0.44$ mg/l and a solution with $C_o = 10,000$ mg/l was injected over a period of $t_o = 1.75$ hours. The dispersion coefficient was constrained to the range 1 - 100 cm²/day based on the information available in Figure 1 and Equation 3 from Gelhar et al. (1992). The retardation factor was approximated as $R = 1.56$ (Korom, 2000(Korom, 2000)).

Finally, arrival time ($t = t_a$) for each sampler was determined by linear interpolation between the first breakthrough above background level ($C_i$). These times and the parameters specified above were used to approximate linear velocities and bulk hydraulic conductivity values for each polygon.

Consistent with the experiment, the arrival time for a sampler located at a distance $x = L$ is significantly larger than the duration of tracer application, $t_a \gg t_o$, warranting the approximation $t_a - t_0 \approx t_a$ and reducing Eq. (1) to:

$$C(x = L, t = t_a) \approx C_i - C_i A(x = L, t = t_a) \tag{3}$$

substituting Eq. (2) and rearranging, we obtain:

$$erfc\left[\frac{RL - vt_a}{2(DRt_a)^{0.5}}\right] + exp\left(\frac{vL}{D}\right) erfc\left[\frac{RL + vt_a}{2(DRt_a)^{0.5}}\right] - 2\left(\frac{C_i - C(L, t_a)}{C_i}\right) = 0 \tag{4}$$

Then, the velocity $v$ is estimated as the root of Eq. (4) (see Table 2). Note that other approaches based on the center of mass of the breakthrough curve have been previously proposed (Feyen et al., 2003; Harvey and Gorelick, 1995; Mercado, 1967); however, they cannot be used in our experiment because only a small fraction of the tracer was recovered after two years of monitoring and the complete flushing of the tracer is likely to take several more.

Bulk hydraulic conductivities were then estimated using the fitted velocity values (Table 2). Vertical velocities were
estimated by substituting $L$ for the depth of the sampler. Horizontal velocities were estimated by substituting $L$ for the shortest horizontal distance from the samplers in rims or troughs to the area of tracer application in the polygon center. When estimating horizontal velocities, vertical arrival times could not be separated from the horizontal arrival times. Thus, resultant estimates for horizontal hydraulic conductivity were low bounding estimates.

Darcy's law was used to constrain hydraulic conductivity


$$K_h = \frac{v_h \theta L_h}{\Delta h} \tag{5}$$

where $K_h$ is the horizontal estimate of hydraulic conductivity, $v_h$ is the horizontal estimate of pore velocity, $\theta$ is effective porosity, $L_h$ is horizontal distance, and $\Delta h$ is the hydraulic head change. Ranges of $K_h$ for each polygon were estimated using maximum and minimum estimated horizontal velocities and the minimum and maximum values of mineral layer porosity reported in the literature,
0.19 and 0.55, respectively (Beringer et al., 2001; Hinzman et al., 1998; Lawrence and Slater, 2008; Letts et al., 2000; Nicolsky et





al., 2009; O'Donnell et al., 2009; Price et al., 2008; Quinton et al., 2000; Zhang et al., 2010). The change in hydraulic head was estimated by finding the average head difference between the well in the polygon center to the well nearest the sampler of interest.

**2.9 Mass Balance**

While it was not possible to close the mass balance of the tracer without compromising the experiment (i.e., by coring or digging
pits) an attempt was made to bracket the mass balance. For each polygon, the largest and smallest breakthrough curves via horizontal flux were used to estimate the limits at which horizontal flux had redistributed tracer from polygon centers to rims and troughs. Only one breakthrough curve was used for the high-center polygon since breakthrough was only detected at one location outside the polygon center. First, each polygon was idealized as a cylinder and its area calculated so flux could be estimated. The radius used for the idealized cylinder of the low-centered polygon was 13.4 m and the radius used for the high-centered polygon
was 3.8 m. Second, flux was calculated as the product of the area, porosity, and velocity. Mineral layer porosity values of 0.19 and 0.55 were used, representing the minimum and maximum porosity values listed above, and the values used for velocity were the previously mentioned linear velocity values. Lastly, the product of the flux and change in tracer concentration over time were integrated with respect to time.

**3 Results**

**3.1 Flow Characteristics (observations)**

From the beginning of July up to the mid-August of each thaw season (2015 and 2016), there was relatively little precipitation in the study area and the water table was receding (Fig. 6 and Fig. 7). In both thaw seasons, most of the precipitation occurred between mid-July and the end of August, concurrent with a rising water table. This behavior is explained by evapotranspiration dominance during the first half of each season, previously observed by (Raz-Yaseef et al., 2017), and infiltration dominance in the
second half. Daily average temperatures fluctuated between -0.7 °C and 7.8 °C and -1.7 °C and 13.7 °C in the 2015 and 2016 thaw seasons, respectively (NOAA-CRN 2015-2016). We observed that the study site was much drier in the 2016 thaw season, with far less standing water in the study area than in the 2015 thaw season. In addition, overland flow was never observed as a result of precipitation events, suggesting high infiltration capacity and dominance of subsurface flow.

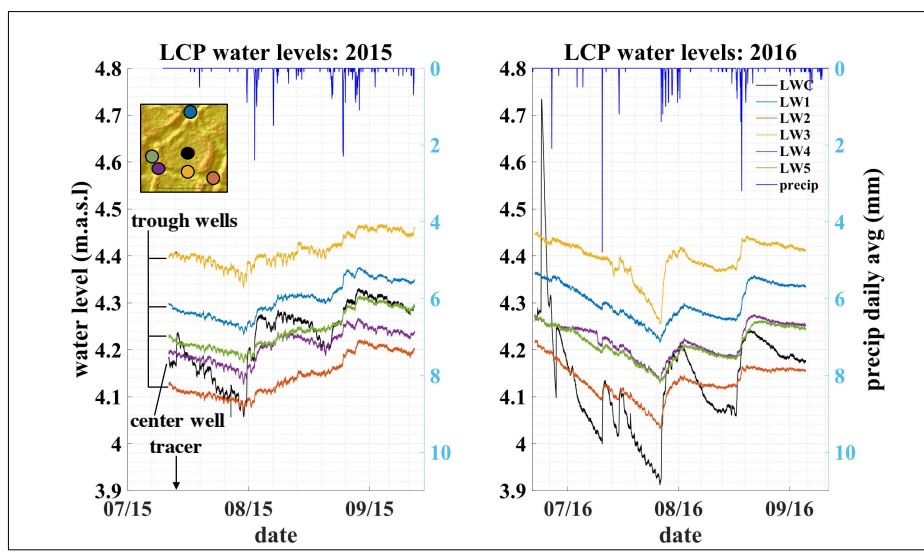




**Figure 6. Water levels from the low-centered polygon (LCP) for the 2015 (left) and 2016 (right) thaw seasons. Arrow indicates date of tracer application. Dots in the inset (upper-left) correspond to observation-well locations; colors of the dots correspond to well hydrographs. Dark blue lines along the top of the graphs indicate hourly total precipitation.**

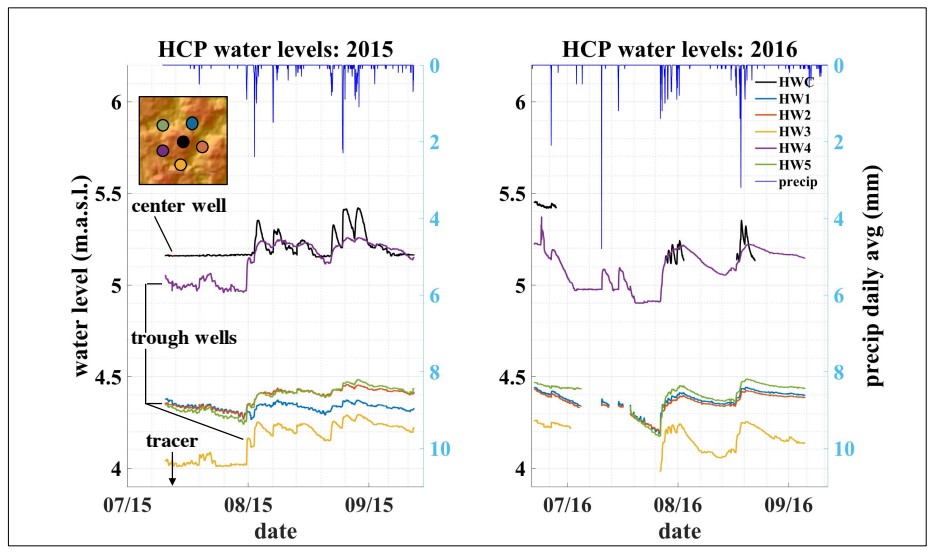

**Figure 7. Water levels from the high-centered polygon for the 2015 (left) and 2016 (right) thaw seasons. Dots in the inset (upper-left) correspond to observation-well locations; colors of the dots correspond to well hydrographs. Dark blue lines along the top of the graphs indicate hourly total precipitation.**

Water levels in the low-centered polygon varied less than 40 cm throughout both thaw seasons with the exception of a

peak in the center well in 2016 (Fig. 6) which is most likely from melt water filling the well when the frost table was only a few centimeters deep. For much of the 2015 thaw season, the water level in the center well was as high as or higher than three out of five of the trough wells. Thus, at the polygon scale, hydraulic gradients were often from the center outward. Conversely, for most of the 2016 thaw season, the water level in the center well was lower than the wells in the troughs, indicating the possibility of the reversal in direction of hydraulic gradient from the polygon center to troughs. Inspection of the well hydrographs reveals that the

center well responded as quickly as trough wells to precipitation events, but with faster increases in water table elevation and steeper recession limbs.

At the high-centered polygon, water levels between center and trough locations often varied by nearly a meter. The center well, HWC, was often dry in the 2016 thaw season as were trough wells HW1, HW2, HW3, and HW5 (Fig. 7). When the center well was not dry, the water level was typically higher than that of the trough wells indicating a hydraulic gradient from the polygon

center to the troughs when water was present. Inspection of the well hydrographs reveals that the well in the polygon center, HWC, had steeper post-precipitation recession limbs than wells in the troughs.

In the low-centered polygon, most precipitation events resulted in higher $\Delta h$ values in the polygon center than in trough wells while characteristic recession times, $\lambda$, in the polygon center were generally shorter than those of trough wells (Table 1). This is consistent with infiltration and ponding in the center of the polygon with subsequent subsurface horizontal redistribution

of mounded groundwater to the troughs. When the system is low in storage (i.e., events 2-4) (Fig. 5 and Table 1), $T_{peak}$ values tend to be lower in trough wells than in the center well. Wells located in the troughs are in topographic lows acting as convergence areas where ponding is likely to occur (Fig. 3). Conversely, when the system is higher in storage (i.e., events 7 and 8), ponding occurs more quickly in the polygon center resulting in $T_{peak}$ values in the polygon center that are shorter or more similar to those in the troughs. This behaviour highlights the importance of microtopography and storage on flow within the low-centered polygon.



The high-centered polygon most often had higher Δh values in the polygon center than in the troughs. Additionally, recession times, λ, were usually shortest in the polygon center with longer times in the troughs. As in the low-centered polygon, this is consistent with groundwater mounding in the center with subsequent subsurface horizontal redistribution to the troughs. Of the trough wells, HW3 and HW4 tended to have the highest Δh values with shorter recession times, λ, than other trough wells (Fig.

7 and Table 1). Notice that these wells are located in high sections relative to the rest of the trough, but low relative to the polygon center. Further, the GPR survey of the high-centered polygon shows that, unlike surface topography, the frost table of the polygon center (the area within the tracer application zone outlined by the blue circle) slopes to the south-southwest (Fig. 8b). This indicates that mounded groundwater from the polygon center is preferentially redistributed to the south-southwest trough of the polygon at HW3 and HW4. Subsequently, mounded water at HW3 and HW4 is redistributed to other parts of the polygon trough. Conversely,

HW1, HW2, and HW5 usually have smaller Δh, shorter times-to-peak, and longer recession times indicating that ponding is dominant in these areas (Table 1).

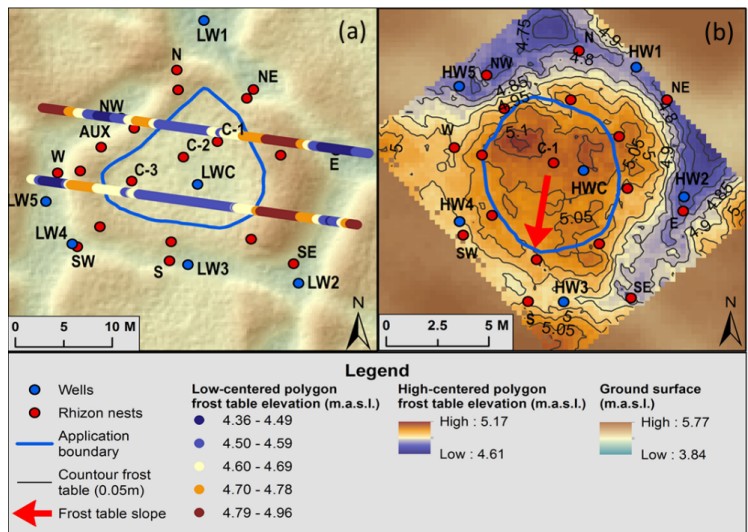

**Figure 8. Frost table elevation obtained with a ground penetrating radar survey at the (a) the low-centered and (b) high-centered polygon locations. Red arrow indicates the direction of frost table slope in high-centered polygon. Widths of transects on low-centered polygon**
**exaggerated for legibility.**

### 3.2 Transport Characteristics (observations)

Tracer breakthrough for both polygons did not exhibit smooth breakthrough curves typically seen in laboratory tracer experiments. Instead, these breakthrough curves have a more jagged form, showing sudden changes in concentration. This jagged nature is due

partly to sampling frequency. Often, there were several days between sampling events resulting in breakthrough curves with a low temporal resolution. However, there is also evidence that large rain events were responsible for some of this variability over time. For example, in the low-centered polygon, there was a concentration increase in well C-1 at the frost table after precipitation Event 3 (Figs. 5 and 9).





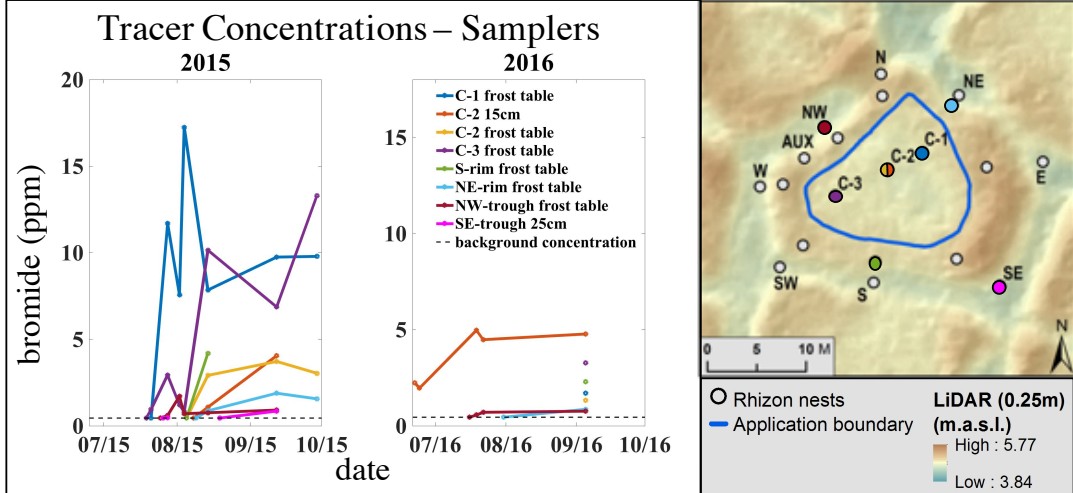

**Figure 9. Tracer breakthrough curves for the low-centered polygon (left) and dots representing their corresponding locations (right). The color of the dots corresponds to the breakthrough curve of the same color. The blue line on the polygon DEM (right) depicts the area of tracer application.**

At the low-centered polygon, tracer arrived first at the center samplers, second in the trough samplers, and third at the rim samplers (Fig. 9 and Table 2). Tracer breakthrough in the center had higher concentrations than in rims or troughs. This was expected as the center was the area of tracer application. While the succession of vertical breakthrough was not entirely captured at all three depths for all three center sampling locations in the low-centered polygon, tracer arrival times were different for each

sampling location in the center. Specifically, tracer arrived first at the frost table of the C-3 sampler after six days, next tracer arrived at the frost table of the C-1 sampler after eight days, at the frost table of the C-2 sampler after 23 days, and finally at the 15 cm depth of the C-2 sampler after 24 days (Fig. 9 and Table 2). Linear velocities of vertical infiltration, calculated using the Eq. (4), varied from 1.55 cm day$^{-1}$ at the 15 cm depth of the C-2 sampler to 30.3 cm day$^{-1}$ at the frost line of the C-3 sampler (Table 2).

The tracer reached the northeast and southern rim locations and the northwest and southeast trough locations of the low-centered polygon (i.e., sampling outside the polygon center) via subsurface flow paths (Fig. 9). As previously mentioned, no overland flow was observed from polygon center to troughs. Interestingly, when tracer breakthrough was detected at trough locations, there was no breakthrough detected at their adjacent rim locations. In addition, all tracer breakthroughs in the rims and the troughs were detected at frost line depth except for the southeast trough location, which occurred at the 25 cm depth, This

highlights the influence of the frost table topography.

Tracer was detected at the more distal trough locations of the low-centered polygon before it was detected at the relatively proximal rim locations. More specifically, tracer arrived first at the two trough locations after 11 and 13 days then at the two rim locations after 23 and 26 days. At the northwest and southeast trough locations, estimated horizontal linear velocities ranged from 17.1-31.3 cm day$^{-1}$ and 38.5-51.8 cm day$^{-1}$, respectively. At the northeast and southern rim locations, estimated horizontal linear

velocities ranged from 6.53-16.5 cm day$^{-1}$ and 8.53-19.5 cm day$^{-1}$, respectively (Fig. 9 and Table 2). Using Eq. (5), the range of horizontal hydraulic conductivity based on first arrivals was estimated to be between $7.67\times10^{-6}$ m s$^{-1}$ and $9.72\times10^{-4}$ m s$^{-1}$.

Tracer was also detected in surface water sampled from the troughs around the low-centered polygon. Samples collected during 2015 show an increasing trend in tracer concentration near the end of the season (Fig. 10b). Even though surface water tracer concentrations for 2015 were relatively low, several of the concentrations were above the 0.44 mg l$^{-1}$ background level for





bromide. This observation can be interpreted as an integrated, well-mixed response of all the tracer that was transported from the polygon center to the troughs via the subsurface. Surface water samples collected during 2016 did not show a clear trend of increasing tracer concentration (Fig 10a).

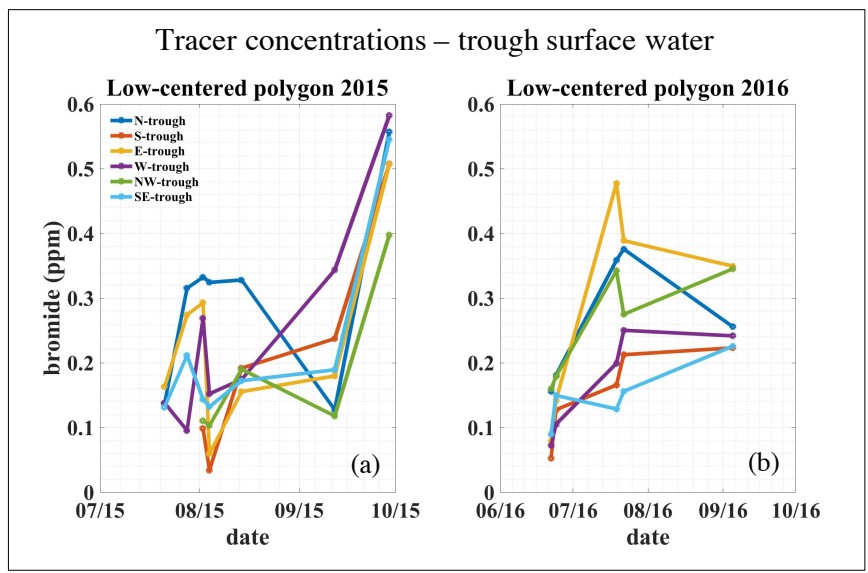

**Figure 10. Breakthrough curves sampled from the surface waters in polygon troughs for (a) low-centered polygon in 2015, and (b) low-centered polygon in 2016. Notice the upward trend in tracer concentration in the troughs of the low-centered polygon (a) during the 2015 thaw season.**

At the high-centered polygon, tracer arrived first in the center (C-1) via vertical infiltration, and second in one rim location via subsurface horizontal flux (Fig. 11 and Table 2). The center location exhibited tracer arrival times that did not necessarily correlate with depth. That is, tracer first arrived simultaneously at the 15 cm and frost table samplers after 21 days, and second at the 25 cm sampler after 34 days. Linear velocities of vertical infiltration, calculated from arrival times, varied from 0.03 cm day$^{-1}$ to 8.3 cm day$^{-1}$.





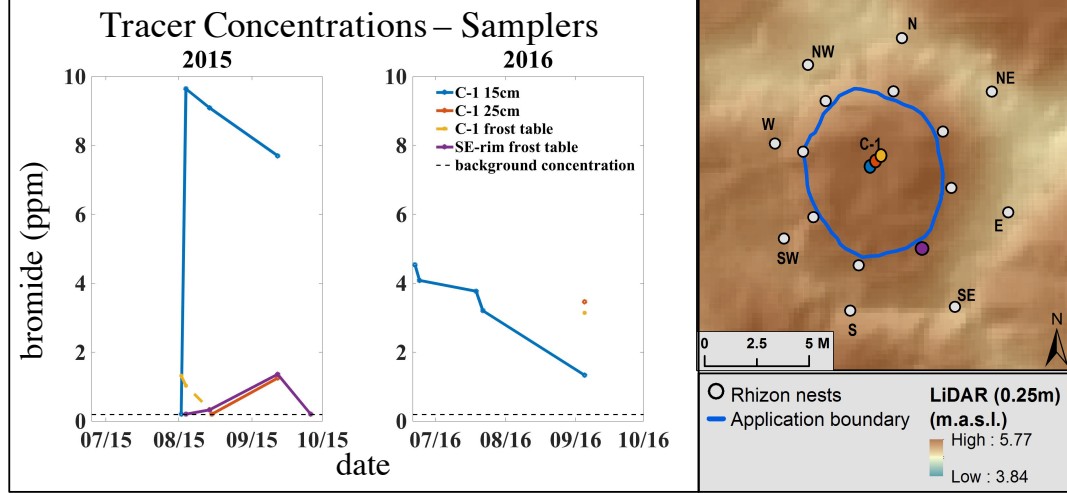

**Figure 11. Tracer breakthrough curves for the high-centered polygon (left) and dots representing their corresponding locations (right). The color of the dots correspond to the breakthrough curve of the same color. The blue line on the polygon DEM (right) depicts the area of tracer application.**

Horizontal tracer flux was evident in only one location outside the center of the high-centered polygon: the southeast rim location (Fig. 11 and Table 2). Tracer arrived at the frost table depth of this location after 23 days. Linear velocity, estimated using Eq (4), was between 0.93 and 9.2 cm/day. The range of horizontal hydraulic conductivity for the high-center polygon was estimated to be between $1.27 \times 10^{-7}$ m s$^{-1}$ and $3.65 \times 10^{-6}$ m s$^{-1}$. Overall, bromide concentrations in trough surface waters of the

high-centered polygon were low. While there were only slight concentration increases (around 0.1 mg l$^{-1}$) in three trough locations at the end of 2015, these concentrations indicate a very small breakthrough, if any.

While horizontal hydraulic conductivity estimates were higher for the low-centered polygon than for the high-centered polygon, there was only one estimate for the high-centered polygon. Unlike the low-centered polygon where tracer was applied to a relatively saturated surface, tracer was applied to a dry surface in the high-centered polygon. In both polygons, estimates of

horizontal hydraulic conductivity are minimum estimates because vertical arrival times could not be separated from horizontal arrival times.

**3.3 Mass Balance**

For the low-centered polygon, the largest tracer mass that could have left the polygon center was estimated to be 93.68% of the tracer. This number is unrealistically high given that the breakthrough curve used in the estimate was incomplete and that the high

bounding value used for mineral porosity was likely overestimated. The smallest tracer mass estimated to have left the center, based on the smallest breakthrough curve, was 4.80%. This number can be considered a "maximum-minimum" and is likely an overestimate since tracer was not detected at all sampling locations around the polygon. For the high-centered polygon, the largest tracer mass that could have left the polygon center was estimated to be 6.82% while the smallest estimated mass was 2.36%. Again, this number can be considered a "maximum-minimum" since the tracer was not detected at all sampling locations around the

polygon. Even though these estimates have large uncertainties, it appears that most of the tracer remains in the center of both polygons.





## 4 Discussion

The general pattern in tracer dynamic in both polygon types was to first infiltrate vertically until it encountered the frost table, then to be transported horizontally, highlighting the influence of the frost table on horizontal flux. There were no new tracer arrival locations during the 2016 thaw season beyond those identified in 2015. Only a small percentage of tracer mass was recovered after monitoring breakthrough over two years, indicating that subsurface flow and transport within both polygon types was very slow. Ranges of hydraulic conductivity estimated for both polygons fall within the range of vertical conductivity found in literature (Atchley et al., 2015; Beringer et al., 2001; Hinzman et al., 1991, 1998; Lawrence and Slater, 2008; Nicolsky et al., 2009; O'Donnell et al., 2009; Price et al., 2008; Quinton et al., 2000; Zhang et al., 2010). However, assuming uniform values for horizontal conductivity is probably inappropriate as there were several rim and trough locations where no tracer was detected.

Overall, the low-centered polygon had higher fluxes and tracer breakthroughs at more locations than in the high-center polygon. The observed differences in tracer flux between polygon types is, in part, explained by degree of saturation. The high-centered polygon, by its very nature, had a higher center relative to the water table and was drier than the center of the low-centered polygon. As a result, the tracer was applied to a more saturated surface on the low-centered polygon as opposed to a dry surface on the high-centered polygon. This allowed the tracer to become mobile more quickly in the low-centered polygon even though the elevation gradient was higher in the high-centered polygon.

### 4.1 Heterogeneity of Flux and Contributing Factors

Both polygon tests demonstrate heterogeneity of vertical and horizontal tracer flux. Preferential flow paths or heterogeneity of subsurface media likely contribute to the heterogeneity of tracer transport observed in both polygon types. Evidence from cores, GPR data, and previous studies provides insight into factors that may be contributing to the heterogeneity of the flow system. Relative importance of these factors still needs to be established.

Results suggest heterogeneity in porous media characteristics affecting vertical transport. Different arrival times at the frost table were observed in the low-centered polygon and various linear velocities were observed in the center of the high-centered polygon (Figs. 9, 11 and Table 2). This indicates preferential flow in the vertical direction and the existence of secondary porosity.

The wide range in horizontal conductivity observed in both polygons is also characteristic of preferential flow paths or heterogeneous subsurface media. Within the low-centered polygon, tracer arrival in the two trough locations was not preceded by tracer arrival in adjacent rim locations (Fig. 9, Table 2). It seems that tracer was able to move to the trough through areas in between the rim sampler locations, at least in the early part of the experiment. For example, in the low-centered polygon, tracer arrived at the northeast trough location, but without ever arriving at the northeast rim location. This implies that flow paths exist that routed the tracer flux around the adjacent rim location to the corresponding trough location. In the high-centered polygon, the only breakthrough detected outside the polygon center was at the southeast rim location (Fig. 11), indicating heterogeneity in horizontal transport.

Characteristics of active layer soils help to explain heterogeneity of flux. For example, variability in vertical and horizontal flux is consistent with the soil structures observed in CT scans of cores taken from other ice-wedge polygons near the study area (Fig. 12a and b). Varying density and patterns of vertical and horizontal density contrasts throughout the cores indicate the potential for preferential flow. Cryoturbation, a freeze-thaw processes that mixes organic and mineral soils within the active layer (Bockheim et al., 1998; Michaelson et al., 1996), is also a potential cause heterogeneity in subsurface media. Cryoturbation results in discontinuous, non-stratified soil horizons. Contrasts in hydraulic properties between discontinuously distributed soil types likely contributes to heterogeneity in flow and transport.



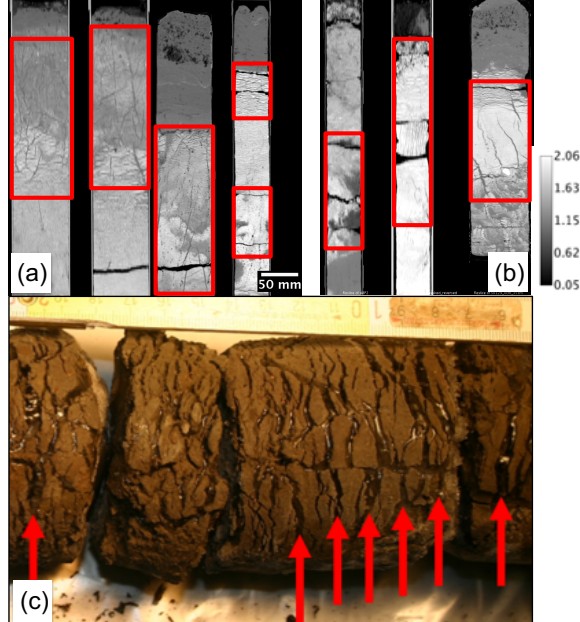

**Figure 12. Vertical cross sections of X-ray CT scans of the top 40 cm of cores from (a) low-centered polygons and (b) high-centered polygons. The CT scans show the density distribution from low (dark) to high (white) with a calibration bar shown in g/cm³. Red boxes indicate patterns of vertical and horizontal density contrasts. Frozen core sampled from saturated tundra (c) is analogous to a low-centered polygon. Red arrows indicate prominent ice lenses. Ice lenses are formed at freeze-up when the soil is sufficiently saturated. Notice that most ice lenses are horizontal relative to ground surface. Photo credit: Vladimir Romanovsky.**

The influence of frost table topography on horizontal flux is significant within both polygons. Outside of polygon centers, tracer was detected almost exclusively at the frost table. The structure of frost table topography can be seen through the GPR survey. The GPR transects at the low-centered polygon show the trend that frost table topography generally follows surface topography (Fig. 8). More specifically, higher-elevation frost table areas are overlain by higher-elevation surface topography and lower-elevation frost table areas are overlain by lower-elevation surface topography. Three of the four locations where tracer was detected outside the center of the low-centered polygon are where the surface topography, and therefore the frost table topography, is relatively low in the rim separating the polygon center and trough (Fig. 8 and Fig. 9). Low points in the topography of the frost table help to explain why breakthrough was detected in these locations. Similarly, in the high-centered polygon, frost table topography within the application area slopes to the south-southwest and the only tracer breakthrough detected outside the polygon center was in the southern half of the polygon (Fig. 8 and Fig. 11).

The presence of ice lenses may also drive differences in subsurface horizontal flux between the low- and high-centered polygons. A frozen core, shown in Fig. 12c, was collected from saturated tundra at the Barrow Environmental Observatory. Although this core was not taken from the polygons used in this experiment, it can be used as an analog for understanding the effect of ice lenses in subsurface structure. In fully saturated tundra, such as the low-centered polygon, ice lenses tend to form during freeze-up whereas they are not as common where tundra is unsaturated, as in the high-centered polygon. This core, taken from a saturated area, exhibits ice lenses up to 3 mm wide primarily in the horizontal plane. We speculate that, as the frost table progressively deepens each year and these ice lenses thaw, some of the resultant cracks remain open enough to create secondary porosity within the low-centered polygon. A system of secondary porosity, oriented primarily in the horizontal plane, helps explain why the low-centered polygon would exhibit faster tracer breakthrough in more rim and trough locations and at higher rates than the high-centered polygon. Whether or not these cracks stay open, and for how long, may be a function of soil structure. For



example, cracks running through soil containing high concentrations of decomposed organic matter may collapse more quickly following thaw of ice lenses while cracks running through soil without decomposed organic matter remain longer. Variable collapse of secondary porosity structures across the polygon may help to explain heterogeneity of horizontal tracer breakthrough in rims and troughs. Furthermore, thicker and more numerous ice lenses tend to form at the bottom of the active layer, near the

top of the permafrost (Guodong, 1983), helping to explain why most breakthrough was observed at the frost table. Additional research is needed to evaluate the importance of ice lenses on flow and transport within polygon systems.

**4.2 Transition between 2015 Freeze-up and 2016 Thaw**

It might be assumed that, due to freezing, tracer migration would resume in 2016 where it ended in 2015. However, in 2016, results show a substantial reduction in tracer concentration as compared to the end of the 2015 thaw season (Figs. 9-11). In fact,

concentrations dropped by as much as 81% and many locations with tracer breakthrough in 2015 did not experience tracer breakthrough until late in the 2016 thaw season. Since tracer was mostly detected at frost table depth in 2015, it would have been contained in the frozen subsurface at the start of the 2016 thaw season. This implies that tracer would not become mobile in the second year until the active layer thawed to the depth at which the tracer was frozen in the first year. Even when the ground thawed to depth of the tracer, the tracer could have been further diluted. That is, once the part of the soil profile containing tracer begins

to thaw it could have also mixed with precipitation that had newly entered the system even before all the tracer-containing ground was thawed. Mixing with precipitation from the second year would cause the tracer to become further diluted and dampen the breakthrough response.

      Redistribution of tracer in the soil horizon during freeze-up may have also been a factor in the reduction of tracer concentrations between the 2015 and 2016 thaw seasons. Tracer freeze-out could have contributed to tracer redistribution between

thaw seasons. Freeze-out is a process by which, as freeze-up progresses, most of the tracer remains in the aqueous component. In the Arctic, the active layer freezes from the top down and the bottom up simultaneously (although not necessarily at the same rate) (Cable, 2016). Thus, the tracer could have been redistributed within the soil profile as a result of freeze-out while remaining mobile in the unfrozen portion of the soil profile until freeze-up was complete. It has also been established that temperature gradients have the potential to cause redistribution of soil moisture (Hinzman et al., 1991; Painter, 2011; Schuh et al., 2017). During freeze-

up, soil moisture in the active layer migrates toward freezing fronts (top and bottom) in a process known as cryosuction. The freeze-out and cryosuction processes could have a combined effect on redistribution of the tracer within the active layer of the polygons.

      Snowmelt at the beginning of the 2016 thaw season may have enabled the rapid removal of some tracer, especially from troughs. No samples were taken during snowmelt which typically only lasts between two and three weeks. A significant reduction

in tracer concentration can be seen in the troughs of the low-centered polygon from the end of the 2015 thaw season to the beginning of the 2016 thaw season (Fig. 10 a and b). Since these represent surface waters, the reduction in concentration here may be explained by snowmelt dilution and runoff. This phenomenon may not affect the subsurface because only a shallow portion of the soil profile is thawed during snowmelt.

**5 Conceptual Model**

Overall, the results suggest a conceptual model of how solutes are transported within ice-wedge polygons (Fig. 13). One early hypothesis was that a contrast in hydraulic properties between the organic and mineral layers would be the primary factor limiting vertical flux and promote lateral flux, which results did not corroborate. In both polygons, the role of the frost table proved to be more important in inhibiting vertical flux as tracer was transported vertically, then horizontally upon encountering the frost table. Results from GPR and well responses indicate that frost table topography also influences the heterogeneous horizontal distribution



of solutes within polygon systems. Horizontal conductivity estimates suggest that, at the polygon scale, horizontal flux does play a role, although perhaps not to the same degree as vertical flux. Both polygons experienced breakthrough in the first thaw season, first in polygon centers via vertical infiltration, then outside of polygon centers at frost table depth via subsurface horizontal flux. Only a small percentage of tracer mass was recovered over two years of monitoring, indicating that flow and transport were very

slow and that both polygons had a high residence time with most of the tracer mass likely remaining in polygon centers.

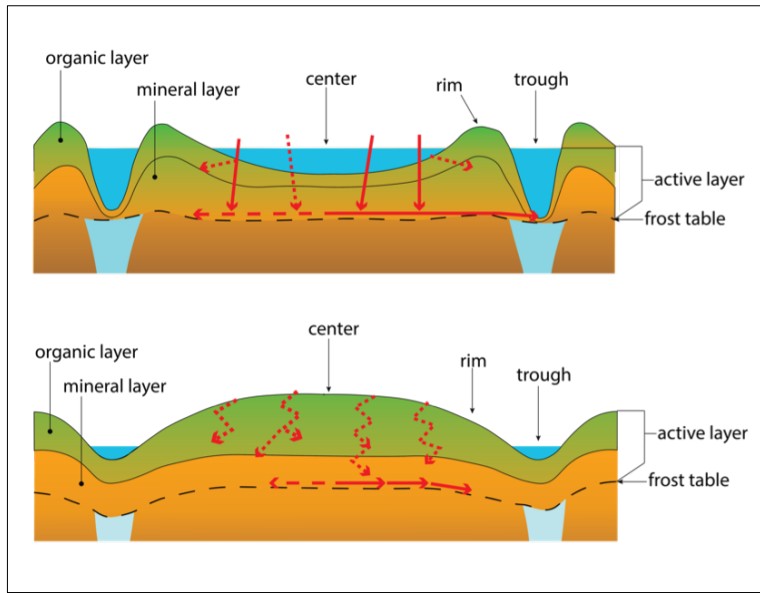

**Figure 13. Conceptual diagram of tracer transport in ice-wedge polygons. Red arrows indicate transport pathways. Length of arrow indicates relative flux. Dashed arrows indicate minimal/negligible flux within a polygon.**

Vertical and horizontal flux proved to be highly heterogeneous in both polygon types. Heterogeneity in vertical flux manifested in both polygons as tracer arrival at deeper depths before arriving at relatively shallow depths at a given location. Heterogeneity in subsurface horizontal flux for both polygon types is demonstrated by the fact that tracer arrived at only 4 out of 17 sampler nests outside of the center in the low-centered polygon and only one out of 16 sampler nests outside of the center in

the high-centered polygon. Tracer arrival times and estimates of horizontal conductivity in the low-centered polygon also show significant variability.

Besides the arrival of tracer outside of polygon centers, the existence of subsurface horizontal flux was also supported by the characteristic responses of observation wells within both polygons. Well responses indicated that water from polygon centers was redistributed to polygon troughs after rain events. Also, there were no observations of overland flow during the course of the

entire experiment, even during multi-day rain events. This implies that, in both low- and high-centered polygons, all flow from the center to the troughs took place in the subsurface. In the low-centered polygon, tracer found in surface water of troughs also supports the existence of subsurface horizontal flux.

While subsurface horizontal flux did play a role in both polygon types, results suggest that the low-centered polygon experienced higher subsurface horizontal flux in more locations than the high-centered polygon. Furthermore, estimates of

hydraulic conductivity in the low-centered polygon were orders of magnitude higher than in the high-centered polygon. Degree of saturation may explain why the low-centered polygon experienced higher subsurface horizontal flux. Saturated media probably




allowed for more immediate mobilization of the tracer in the low-centered polygon. The formation of ice lenses in the low-centered polygon is another possible explanation of the increased horizontal conductivity due to secondary porosity.

Even though temporal changes are not shown in Fig. 13, temporal aspects are an important part of the conceptual model. For example, while the changing elevation of the frost table and its topography is not depicted, it plays an important role in inhibiting infiltration and influencing preferential flow. Similarly, the transition between winter freeze up and subsequent thaw also have a significant effect on tracer transport.

This conceptual model has implications for biogeochemical processes. Distinct biogeochemical differences have been shown to exist between different microtopographic features of polygons (Andresen et al., 2016; Lara et al., 2015; Liljedahl et al., 2016; Newman et al., 2015; Wainwright et al., 2017; Zona et al., 2011b). These tracer results suggest there are lateral connections between polygon centers and troughs that can facilitate transport of chemical species across microtopographic features. This transport could affect redox conditions and have multiple effects on biogeochemical cycling. For example, water in the centers of high-center polygons has been shown to have significantly higher concentrations of nitrate and sulfate than water in the troughs (Heikoop et al., 2015; Newman et al., 2015). When water in the center of a high-center polygon is redistributed to the polygon trough, due to preferential pathways, it could cause oxyanions to be transported to the troughs in a heterogeneous manner. In turn, this may influence carbon partitioning by influencing microbial respiration and plant growth (Weintraub and Schimel, 2005; Zona et al., 2011b). As a result, models including horizontal flux may represent carbon partitioning and the migration of plant communities more accurately than models that assume only vertical flux is relevant.

## 6 Conclusions

Our study provides new insight into hydrological processes of low- and high-centered polygon systems, where flow and transport field investigations are almost totally lacking. This study shows that polygon type significantly affects flow and transport as faster and more prolific tracer breakthrough was observed in low-center polygons than high-center polygons. Results suggest that horizontal flow is important, and that heterogeneity of subsurface media plays a significant role in flow and transport. Our study also provides some evidence about what is potentially controlling heterogeneity. These insights can help to improve hydrological models such as the Arctic Terrestrial Simulator (Atchley et al., 2015; Painter et al., 2016).

Given that much of the tracer remains in the polygons, longer-term monitoring would contribute to a better understanding of these systems. Additional work is also needed toward understanding controls on heterogeneity of flux in ice-wedge polygons. For example, the effect of ice lenses and cryoturbation on flux require further investigation.

*Data availability*. Data sets are available on request at the NGEE-Arctic data repository [Wales et al., 2017] (http://dx.doi.org/10.5440/1342954).

*Author contributions*. BDN originally conceived of the experiment and NW and CJW helped to refine the experimental design. NAW performed the majority of field work, data analysis, and wrote the paper. BD performed the GPR survey of the study site. TJK collected frozen cores and conducted CT scans. JDGV provided contributions to design and implementation of the data analysis approach and helped with the writing of some sections, including the results. CJW and SDW provided institutional oversight. All authors provided comments and text prior to the submission of the paper.

*Competing interests*. The authors declare that they have no conflict of interest.



*Acknowledgements.* The Next-Generation Ecosystem Experiments (NGEE Arctic) project is supported by the Office of Biological and Environmental Research in the DOE Office of Science. We are grateful to George Perkins, Emily Kluk, and the staff of the GGRL at Los Alamos National Laboratory for their work in sample analysis. We thank Anna Liljedahl for early contributions to experimental design and for the use of preexisting infrastructure. We also thank Lauren Charsley-Groffman for GIS work and her

work in the field. We thank John Peterson for helping GPR data acquisition and processing. We acknowledge Vladimir Romanovsky for his pictures of frozen cores and Bob Busey for the use of his meteorological data. This work has been authored by an employee of Los Alamos National Security, LLC, operator of the Los Alamos National Laboratory under contract DE-AC52-06NA25396 with the U.S. Department of Energy. The United States Government retains and the publisher, by accepting this work for publication, acknowledges that the United States Government retains a nonexclusive, paid-up, irrevocable, worldwide license

to publish or reproduce this work or allow others to do so for the United States Government purposes.

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





**Tables**

| | | | LCP | | | | | | HCP | | | | | |
|---|---|---|---|---|---|---|---|---|---|---|---|---|---|---|
| | | | LWC | LW1 | LW2 | LW3 | LW4 | LW5 | HWC | HW1 | HW2 | HW3 | HW4 | HW5 |
| **2015** | event 1 | $\Delta h$ [m] | N/A | N/A | N/A | N/A | N/A | N/A | N/A | N/A | N/A | 0.06 | 0.06 | N/A |
| | | $T_{peak}$ [d] | N/A | N/A | N/A | N/A | N/A | N/A | N/A | N/A | N/A | 0.22 | 0.28 | N/A |
| | | $\lambda$ [d] | 357 | 1352 | 2425 | 909 | 1315 | 1552 | N/A | 714 | 714 | 116 | 213 | 417 |
| | event 2 | $\Delta h$ [m] | 0.166 | 0.018 | 0.013 | 0.576 | 0.024 | 0.013 | N/A | 0.018 | 0.06 | 0.15 | 0.183 | 0.098 |
| | | $T_{peak}$ [d] | 0.75 | 0.73 | 0.35 | 0.32 | 0.75 | 0.35 | N/A | 0.32 | 0.78 | 0.40 | 0.79 | 0.80 |
| | | $\lambda$ [d] | 57 | 143 | N/A | 68 | 85 | 91 | N/A | 101 | 76 | 72 | 78 | 101 |
| | event 3 | $\Delta h$ [m] | 0.09 | 0.03 | 0.03 | 0.02 | 0.04 | 0.03 | 0.18 | 0.08 | 0.07 | 0.14 | 0.13 | 0.08 |
| | | $T_{peak}$ [d] | 1.96 | 1.88 | 1.75 | 0.89 | 1.90 | 1.75 | 1.05 | 1.93 | 2.93 | 1.00 | 2.00 | 1.97 |
| | | $\lambda$ [d] | 333 | 84854 | 19846 | 667 | N/A | N/A | 70 | 714 | 1313 | 204 | 400 | 556 |
| | event 4 | $\Delta h$ [m] | 0.03 | 0.03 | 0.02 | N/A | 0.02 | 0.02 | 0.13 | 0.03 | 0.03 | 0.08 | 0.04 | 0.03 |
| | | $T_{peak}$ [d] | 1.32 | 1.30 | 1.30 | N/A | 1.30 | 1.30 | 1.42 | 1.28 | 1.29 | 1.40 | 1.32 | 1.30 |
| | | $\lambda$ [d] | N/A | N/A | N/A | N/A | N/A | N/A | 66 | 588 | 500 | 286 | 909 | 714 |
| | event 5 | $\Delta h$ [m] | N/A | N/A | 0.01 | 0.02 | N/A | N/A | 0.08 | N/A | 0.02 | 0.05 | 0.02 | N/A |
| | | $T_{peak}$ [d] | N/A | N/A | 0.64 | 0.66 | N/A | N/A | 1.72 | N/A | 0.46 | 2.11 | 2.65 | N/A |
| | | $\lambda$ [d] | 286 | N/A | N/A | 909 | 1043 | N/A | 87 | 909 | 500 | 233 | 244 | N/A |
| | event 6 | $\Delta h$ [m] | 0.05 | 0.03 | 0.01 | 0.01 | 0.02 | 0.01 | 0.17 | 0.03 | 0.03 | 0.11 | 0.10 | 0.04 |
| | | $T_{peak}$ [d] | 1.75 | 2.95 | 0.96 | 1.34 | 1.00 | 0.95 | 1.13 | 1.02 | 1.05 | 1.23 | 2.70 | 1.42 |
| | | $\lambda$ [d] | N/A | N/A | N/A | N/A | N/A | N/A | 66 | 714 | 2345 | 204 | N/A | N/A |
| | event 7 | $\Delta h$ [m] | 0.05 | 0.04 | 0.03 | 0.02 | 0.03 | 0.03 | 0.22 | 0.04 | 0.03 | 0.06 | 0.05 | 0.03 |
| | | $T_{peak}$ [d] | 1.28 | 1.19 | 1.38 | 1.25 | 1.20 | 1.38 | 0.38 | 0.34 | 0.33 | 0.46 | 1.36 | 0.35 |
| | | $\lambda$ [d] | 1532 | 556 | 1219 | 556 | 588 | 1000 | 70 | 357 | 385 | 244 | 417 | 556 |
| | event 8 | $\Delta h$ [m] | 0.03 | 0.02 | 0.03 | 0.02 | 0.02 | 0.03 | 0.17 | 0.03 | 0.02 | 0.04 | 0.02 | 0.03 |
| | | $T_{peak}$ [d] | 0.86 | 1.17 | 1.18 | 1.07 | 0.84 | 1.18 | 0.85 | 0.75 | 0.81 | 0.89 | 1.08 | 0.78 |
| | | $\lambda$ [d] | 769 | 714 | 1687 | 1827 | 667 | 909 | 57 | 455 | 526 | 333 | 455 | 476 |
| **2016** | event 9 | $\Delta h$ [m] | N/A | 0.01 | N/A | 0.01 | N/A | N/A | 0.02 | 0.02 | 0.02 | 0.03 | 0.05 | N/A |
| | | $T_{peak}$ [d] | N/A | 0.42 | N/A | 0.34 | N/A | N/A | 0.31 | 0.39 | 0.34 | 0.34 | 0.74 | N/A |
| | | $\lambda$ [d] | 29 | 833 | 625 | 1528 | 2851 | 1324 | 455 | 435 | 417 | 714 | 130 | N/A |
| | event 10 | $\Delta h$ [m] | 0.13 | 0.01 | 0.03 | N/A | 0.02 | 0.01 | N/A | 0.03 | 0.01 | N/A | 0.11 | N/A |
| | | $T_{peak}$ [d] | 0.79 | 0.84 | 0.85 | N/A | 0.83 | 1.08 | N/A | 0.20 | 0.20 | N/A | 0.77 | N/A |
| | | $\lambda$ [d] | 115 | 625 | 1134 | 2220 | 833 | 667 | N/A | 345 | 556 | N/A | 114 | N/A |
| | event 11 | $\Delta h$ [m] | 0.10 | 0.01 | 0.02 | 0.03 | 0.01 | 0.01 | N/A | 0.01 | N/A | N/A | 0.12 | N/A |
| | | $T_{peak}$ [d] | 0.41 | 1.24 | 1.23 | 0.36 | 1.12 | 1.27 | N/A | 0.26 | N/A | N/A | 0.44 | N/A |
| | | $\lambda$ [d] | 250 | 556 | 455 | 244 | 556 | 1000 | N/A | 152 | 286 | N/A | 118 | N/A |
| | event 12 | $\Delta h$ [m] | N/A | N/A | N/A | N/A | N/A | N/A | N/A | N/A | N/A | N/A | N/A | N/A |
| | | $T_{peak}$ [d] | N/A | N/A | N/A | N/A | N/A | N/A | N/A | N/A | N/A | N/A | N/A | N/A |
| | | $\lambda$ [d] | 303 | N/A | 556 | 385 | 769 | 909 | N/A | 333 | 417 | N/A | N/A | 270 |
| | event 13 | $\Delta h$ [m] | 0.29 | 0.08 | 0.11 | 0.16 | 0.09 | 0.08 | 0.12 | 0.16 | 0.20 | 0.26 | 0.31 | 0.27 |
| | | $T_{peak}$ [d] | 5.89 | 6.53 | 6.50 | 5.43 | 6.54 | 6.56 | 5.49 | 6.49 | 6.50 | 5.49 | 6.49 | 5.47 |
| | | $\lambda$ [d] | 238 | 1432 | 1553 | 400 | 1320 | 2106 | 53 | 556 | 556 | 156 | 313 | 435 |
| | event 14 | $\Delta h$ [m] | 0.18 | 0.09 | 0.04 | 0.07 | 0.09 | 0.08 | 0.23 | 0.09 | 0.09 | 0.17 | 0.15 | 0.12 |
| | | $T_{peak}$ [d] | 4.07 | 5.14 | 2.96 | 3.00 | 5.06 | 7.17 | 1.75 | 1.85 | 2.98 | 3.00 | 4.16 | 3.34 |
| | | $\lambda$ [d] | 714 | 1720 | 3125 | 1323 | 1857 | 6025 | 52 | 769 | 1000 | 333 | 625 | 769 |

**Table 1. Response and recovery data from observation wells. Shaded with bold font indicates wells in polygon centers. Change in head ($\Delta h$)[m], time to peak $T_{peak}$[days], and characteristic response ($\lambda$)[days]**





| Low-centered polygon | | | |
|---|---|---|---|
| location | distance (cm) | arrival time (days) | min velocity (cm/day) | max velocity (cm/day) |
| C-1* | 36.5 | 8 ± 1 | 2.70 | 20.97 |
| C-2 | 15 | 24 ± 1 | 1.55 | 7.39 |
| C-2* | 61 | 23 ± 1 | 1.60 | 11.23 |
| C-3* | 58 | 6 ± 1 | 7.40 | 30.30 |
| S-rim* | 222 | 23 ± 1 | 8.53 | 19.49 |
| NE-rim* | 197 | 26 ± 1 | 6.53 | 16.56 |
| NW-trough* | 363 | 11 ± 1 | 17.14 | 31.33 |
| SE-trough | 694 | 13 ± 1 | 38.52 | 51.85 |
| High-centered polygon | | | |
| location | distance (cm) | arrival time (days) | min velocity (cm/day) | max velocity (cm/day) |
| C-1 | 15 | 21 ± 1 | 1.64 | 7.10 |
| C-1 | 25 | 34 ± 1 | 0.03 | 5.49 |
| C-1* | 25 | 21 ± 1 | 0.22 | 8.30 |
| SE-rim* | 43.5 | 23 ± 1 | 0.93 | 9.20 |

**Table 2. Tracer breakthrough locations, times, and linear velocities. Red text indicates uncertainty in tracer arrival time due to insufficient data for interpolation. \*denotes samplers at frost table depth. Shaded denotes vertical flow (center wells) and unshaded denotes horizontal flow.**