# Peer review of "Understanding the Relative Importance of Vertical and Horizontal Flow in Ice-Wedge Polygons"

_Hydrology and Earth System Sciences, 2019_

## Referee Comment (RC1) · Anonymous Referee #1 · 24 May 2019

**General comments:**

This paper documents the findings from field observations of subsurface routing in high and low centered polygons in continuous permafrost. The authors used a conservative tracer and hydraulic head measurements from a series of wells to estimate subsurface runoff. The authors claim that most hydrological models do not have processes to represent lateral routing and that this paper demonstrates that this process be included in land surface schemes. For the most part (with the exceptions noted below), the science seems sound, however a mass balance of the bromide tracer was unachievable due to possible cryoturbation or other redistribution processes during freeze-up. I feel that the findings of this paper could merit publication; however there are some very major revisions that are required, including substantial rewriting. As it is written, the paper does adequately place this study in the context of previous research and the results are not clearly defined. The abstract and conclusion need to be re-worked to identify the scientific observations that will benefit the hydrology community.

It is my understanding that the authors are claiming that lateral transport across the frost table after infiltration is the most important finding of their study. The idea that frost table topography controls subsurface runoff has been well documented in the literature (Morison *et al*., 2016, Helbig *et al*., 2013; Quinton *et al*., 2000; Wright *et al*., 2009) and should be acknowledged as such, instead of as a novel finding. It was surprising that the authors briefly cited some very relevant studies for general water balance estimates (*i.e.* Helbig *et al*., 2013 for evapotranspiration; Liljedahl *et al*., 2016 for biogeochemical comparisons; Quinton *et al*., 2000 for hydraulic conductivity), but did not mention these studies in their discussion of subsurface routing in Arctic environments (and specifically ice-wedge polygons). By citing these papers the authors demonstrate that they are aware of these studies, but for some reason do not frame their research in the context of work that has already been completed. In the abstract, the sentence, "Estimates of horizontal hydraulic conductivity were within the range of previous estimates of vertical conductivity, highlighting the importance of horizontal flow in these systems" appears to be the most conclusive sentence in the abstract but does not convince the reader of a novel finding. The main finding in the conclusion is that, "horizontal flow is important". After reading this paper, I have not been convinced that horizontal flow is 'important', nor do I have an idea of how important it is on the total flux of subsurface runoff. I am also not convinced that this study, as-is, will provide a basis to improve hydrological models. In making these claims, the authors should: a) quantify horizontal hydraulic conductivity rates (this could be done directly in the field); and b) identify lateral flow routing mechanisms and attempt to quantify a landscape flux to demonstrate the relevance to this study. To do this, the results and discussion sections should be re-written to better position the paper's objectives and the authors should consider upscaling their findings to the subcatchment scale. The discussion section should be better framed with more reference to existing literature. As currently written, most of the discussion lacks references, with the exception of occasional sentences having many references (*i.e.* page 20, line 8). The discussion section is a major weakness of the paper and could be written much better. Specific comments are listed below.

**Specific comments:**

**Page 1 lines 29-32**: List references after each point instead of at end of the sentence. For example, "… as it affects hydrology (hydrology refs), biogeochemical transformations (biogeochemical refs)" etc.

**Page 1 line 32:** How is 'northern' Arctic permafrost zone defined? All Arctic landscapes are northern, are you referring to the northernmost Arctic landscapes?

**Page 2, lines 19-20**: I struggle to understand the notation of the 'relative' roles of vertical and horizontal fluxes and that no other studies have been conducted toward quantifying this. It is generally accepted that in permafrost environments precipitation inputs: 1) infiltrate organic soils; 2) percolate to the frost table; and 3) produce lateral runoff where it is routed in accordance with frost table topography and is governed by fill-and-spill. There has been considerable work evaluating this principle, and other bodies of work that have evaluated subsurface runoff through ice wedge polygon terrain at the landscape scale (Helbig *et al*., 2013; Liljedahl *et al*., 2016). I am also not convinced that if regional and pan-Arctic land models ignore horizontal fluxes that they would be well positioned to incorporate results of a study that document flow at the individual polygon scale. Furthermore, there are many hydrological models that include modules for subsurface routing, and even have options for different ways to parameterise that routing (*i.e.* Raven hydrological framework, Cold Regions Hydrological Model, Canadian Land Surface Scheme). I think the authors should also stress that this study seeks to better understand the differences in subsurface hydrology between low and high centered polygons, as this is a key component of the research.

**Page 3, line 10:** Again, routing mechanisms for lateral flow in polygonal terrain have been discussed. Helbig *et al*. (2013) conclude, "The prominent microtopography of the polygonal tundra strongly controls lateral flow and storage behaviour".

**Page 4, line 5**: How representative are the properties of the polygons that were selected? Can you provide mean surface area and elevation (DEM?) for the study site?

**Page 5, line 2**: Are the pressure transducers absolute or vented? If the former, where is barometric pressure being collected?

**Page 5, line 2:** How were the elevations of the well casings surveyed?

**Page 5, line 29:** How frequently was the sampler at the frost table moved down?

**Page 8, figure 5:** I would include this in the results section

**Page 9, line 34:** Why were values (and a subsequent range) for porosity used from the literature and not measured at the site?

**Page 10, line 1:** There are numerous instances where this long list of references is used. It would be much more beneficial (and more informative) to include all of these studies in a table

with their associated values for each parameter and then reference that table throughout the paper.

**Page 10, line 2:** What was the time period over which the average head difference was calculated?

**Page 10, line 10:** How was the flux through organic soil calculated?

**Page 10, line 19:** How does 'infiltration dominance' explain a rising water table? These sentences are worded awkwardly. This information would be much more clearly explained by showing a combined plot of cumulative evapotranspiration and cumulative precipitation.

**Pages 10, 11, figures 6, 7:** These are very nice figures and display a lot of data in a format that is easy to read and digest.

**Page 11, line 12:** I do not agree that the 2015 data shows that, "hydraulic gradients were often from the centre outward". I would argue that the hydraulic gradient was variable across the polygon. Also, this was not mentioned in the methods, but how were the wells surveyed and what was the error associated with these surveys? This may impact the hydraulic gradient measurements given that the elevation of all six water tables are within 40 cm.

**Page 11, line 17**: Given their close proximity, why would the purple and yellow trough wells on the HCP have water table differences of nearly one metre? Are there significant differences in soil type, topography, etc?

**Page 15, line 20:** Again, why not measure porosity of the mineral soil directly?

**Page 15, line 21:** A range of 4.8 – 93.7% for possible tracer mass to leave the polygon is very high.

**Page 15, line 25:** Can you conclusively say that 'most' of the tracer remains in the centre of the LCP if your maximum estimate is that 93.7% left? Is there any way to improve this estimate? As-is, you cannot make this claim.

**Page 16, lines 7-8:** Again, provide these references as a table with associated values

**Page 16, line 23:** Can you elaborate on the secondary porosity network and describe this more in Figure 12?

**Page 16, line 33:** It may be worthwhile to include a discussion of heterogeneity and dual porosity in peat as well (I inferred that this section is restricted to the mineral soils).

**Page 17, line 10:** Do the frost table elevations measured with a frost probe coincide with the GPR results?

**Page 17, lines 8-18:** This is a good example of a paragraph that should be linked to existing literature that has evaluated the controls that the frost table exerts on subsurface runoff. A major weakness of this paper is that the discussion section does not integrate this study with other work to advance scientific understanding.

**Page 17, line 23:** "… as the frost table progressively deepens each year and these ice lenses thaw …" – This sentence implies that the active layer is becoming thicker every year. Is this the case? I have not seen a site where the active layer is thicker every year. Also, this section should contain mention of the ice-rich 'transient layer' described by Shur *et al* (2005). This discussion would be strengthened by including different values for hydraulic conductivity as the thawing front transitions from organic to mineral soil, and the controls that soil type has on subsurface runoff.

**Page 18, line 5:** What you are describing here is the transient layer (Shur *et al*, 2005). Again, a more detailed literature review is necessary to better frame the findings from this study.

**Page 18, line 29:** Provide a reference for the statement that snowmelt only lasts between two and three weeks.

**Page 18, section 4.2:** This appears to be a long-winded explanation of why part of the experiment failed, including explanations of various permafrost processes that have been explained before. This section could be greatly reduced and moved to the results section. Was there any monitoring of tracer concentration during freeze-up? This is a period of hydrologic activity that is often overlooked.

**Page 18, line 35:** The initial hypothesis that the interface of organic and mineral layers does not control horizontal flux may still be true. The authors should evaluate the relative roles of the horizontal flux while the frost table is in within the organic layer and when it descends to the mineral soil layer. The effect of subsurface runoff and the interplay between soil layers and frost table dynamics is a process that has been well documented, and should be referenced as such.

**Page 19, figure 13:** In the high centered polygon, why is the vertical flux minimal/negligible? What happens to precipitation inputs if they do not infiltrate the soil column? If this is a conceptual diagram, should the water table in the centre of the polygon (LCP) not be higher than the trough if flow is directed outwards? Why is the major transport pathway to the right and not the left? There does not appear to be a difference in hydraulic gradient. Is this process limited by soil heterogeneity and differences in hydraulic conductivity? The rationale behind this diagram is not clearly evident.

**Page 19, lines 18-19:** Figure 6 does not indicate that water from polygon centres is distributed to troughs in LCPs. Actually, the data from 2016 indicates the opposite (as is stated in the results section). The discussion section should be written to better represent the data.

**Page 19, line 25:** Would estimates of hydraulic conductivity not have been more reliable by completing pump/slug tests in the field?

**Page 20, lines 5 and 6:** Can the impacts of freeze-up and thaw be elaborated? What effect does the two-sided freezing front have on subsurface hydrology in the thawed, saturated zone?

**Page 20, line 20:** I would not agree that field investigations are "almost totally lacking".

**Page 20, line 22:** A major weakness of this study is that the lateral flux is not quantified. Indicating that lateral flow is 'important' is not a conclusion. A total flux (mm) from each polygon is needed if this work is to improve hydrological models.

**Page 20, line 24:** Is the Arctic Terrestrial Simulator the only hydrological model that these insights can help to improve? What is the rationale for including this model?

**Page 20, line 27:** The final sentence is not a good concluding sentence for this paper.

**Technical corrections:**

**Page 1 lines 35-36:** The last two sentences are not sentences. Please rewrite.

**Page 2 line 3:** "centers, rims, and **troughs**" Misspelled.

**Page 10, line 16:** "From the beginning of July **until mid-August**…"

**Page 13, line 20:** "Frost **table** depth"

**Page 16, line 1:** "… tracer dynamic**s** …"

**Page 16, line 21:** First sentence is not a sentence
**Page 16, line 24:** "… range in horizontal **hydraulic** conductivity …"

**Page 16, lines 34 -35:** Awkward sentence

**Page 16, line 35:** "process", not processes

**Page 16, line 36:** "… a potential cause **of** heterogeneity …"

**Additional References:**
Morison, M.Q., M.L. Macrae, R.M. Petrone, and L. Fishback, L (2016), Seasonal dynamics in shallow freshwater pond-peatland hydrochemical interactions in a subarctic permafrost environment, *Hydrological Processes*, 31: 462-475, doi: 10.1002/hyp.11043.
Shur, Y., K.M. Hinkel, and F.E. Nelson (2005), The Transient Layer: Implications for Geocryology and Climate-Change Science, *Permafrost and Periglac. Process.*, 16, 5-17, doi:10.1002/ppp.518.
Wright, N., M. Hayashi, and W.L. Quinton (2009), Spatial and temporal variations in active layer thawing and their implication on runoff generation in peat-covered permafrost terrain, *Water Resources. Research.*, 45, W05414, doi:10.1029/2008WR006880.

---

## Referee Comment (RC2) · Anonymous Referee #2 · 3 Jun 2019

General comment:

The authors report on a bromide tracer experiment that took place in a single high-centered polygon and a single low-centered polygon in northern Alaska at the Barrow NGEE-Arctic site. The tracer was applied in 2015 and then measured through several sampling ports installed at different locations and depths across the polygon, including in adjacent troughs. The field conditions at the site are difficult and the thaw season is short; hence, the amount of data is sparse, as is the potential to conduct similar experiments across a larger number of polygons. The authors used a 1-D analytical solution to the convective-dispersion equation to estimate subsurface flow parameters,

including vertical and lateral hydraulic conductivity (it appears that retardation factor was assumed based on a literature value). The comments below identify a number of areas that need further consideration. For example, the analytical solution assumes a point application, but the tracer in this case was applied to a large area; how should we interpret the boundary conditions used to determine lateral transport parameters? Also, the authors did not include any soil temperature in the manuscript, which would help identify freeze up and thaw, and the potential existence of ice lenses that would almost certainly impact the uniformity of vertical soil water flow. Without these data, the authors relied on conjecture to explain non-uniform transport behavior through the upper thawed soil. It is recommended that the authors include the time-series data on ice table depth, thus potentially helping here. Other comments are found below.

Specific comments – comments called out by x/y, where x is page and y is line number

3/8 – authors should clarify here that only one high-centered polygon and one low-centered polygon were analyzed. As written, it appears that multiple polygons of both types were studied.

4/15 – what was the total area into which bromide tracer was applied?

5/8 – swap Figs. 4a and 4b to follow the order of call outs. Also, the description of the field setup using the silicon sheets doesn't appear on the subfigures. Suggest showing more detail in the schematic, so that the reader can note the silicon sheet, and that "surface" equals ground surface in current Fig. 4b.

5/26 – does the HCP have rims, as indicated in the sentence?

6/18 – given that ponded water apparently existed in the LCP during tracer application, any information on soil water content to confirm that the thawed soil was fully wetted?

6/30 – any soil temperature here or elsewhere at BEO that might be applicable here? Also, it would be helpful for the authors to add a table (here or SI) that lists the frost table depth with time, especially given the importance to lateral transport and heterogeneity

of the frost table depth.

Figure 5 – suggest adding calendar date to either the x-axis or the caption, so that the reader can understand year-to-year variability of onset of thaw

8/10 – van Genuchten and Alves (1982) solution assumes 1D transport, or in the context of this experiment, a point application of tracer. How does the broad area of application square with this assumption? Was it only used to estimate velocities during that segment of the flowpath, and then a second calculation for estimating horizontal flow? How is lateral distance determined for those sampling clusters outside of the application area? Also note that the van Genuchten and Alves reference on 24/33 is incomplete.

9/5 – check table 2. As presented, neither background concentrations nor tracer injection data are included

9/9 – the retardation factor for Korom's experiment were for sediment with a pH of between 5.1 and 5.7. According to Goldberg and Kabengi (2010, doi:10.2136/vzj2010.0028), retardation of bromide is very pH dependent. In some cases, bromide transport in soil with can lead to retardation factors significantly less than one (see for example Hills et al., 1991, WRR, paper 91WR015). How do the soil conditions at the Barrow site compare with those from Korom? Are the data robust enough to estimate R either through parameter estimation or other means? Given how R scales the tracer velocity, so more thought on this issue is warranted.

9/23 – any particular reason why sampling and analyses occurred for only two years, when it became clear that tracer recovery would be so low?

9/25 – here and elsewhere, it is suggested that the authors refer to tracer application in the polygon interior, rather than application in the polygon center. Indeed, most of the interior of the polygon received tracer, rather than a point application.

10/8 – if I understand the narrative correctly, the polygon was represented as an idealized vertical cylinder, and the flux was estimated through the bottom of the cylinder based on measurements from the rhizon nests, is that correct? Was the flux then used as initial conditions for the lateral flow the nests outside of the cylinder?

11 (general) – the authors seem to bounce from LCP and HCP results, first referring to water levels, then to delta H values for both. It would be easier to discuss LCP first, then HCP second

Figure 8 – Fig. 8a shows location of GPR measurements and results, but not frost table slope, and Fig. 8b shows frost table slope but not GPR measurements. Could both results be shown for both polygons?

14/2 – replace "Surface" with "Trough"

14/5 – similar to the comment above, any soil temperature data that could help interpret these results in successive years? The reduced concentration from the end of 2015 to the beginning of 2016 is puzzling and potentially indicates transport even though water appeared frozen.

15/22 – are the authors stating that tracer recovery of 4.80% is actually a high estimate?

15/25 – when authors refer to polygon 'center,' is this really the polygon 'interior?'

16/21 – authors are using either preferential flowpaths or heterogeneity of subsurface media as possible reasons for non-uniform vertical flow, or bypass flow around shallow samplers. A third explanation here is that the soil has undergone partial melting or partial freezing, reducing liquid water-filled transport pathways, and facilitating transport through specific pathways. This might also explain why tracers are changing concentration so drastically between thaw seasons.

Figure 12 – though the figures are interesting, there's not enough explanation behind them to know whether the conditions represented by these images are the same as those observed at the traced polygons. It is suggested that the authors either more

closely tie the images from Romanovsky to the site being reported on here, or consider removing the figures altogether.

---

## Author Comment (AC1) · 26 Jul 2019

**Response to Referee #1**

**General comments:**

This paper documents the findings from field observations of subsurface routing in high and low centered polygons in continuous permafrost. The authors used a conservative tracer and hydraulic head measurements from a series of wells to estimate subsurface runoff. The authors claim that most hydrological models do not have processes to represent lateral routing and that this paper demonstrates that this process be included in land surface schemes. For the most part (with the exceptions noted below), the science seems sound, however a mass balance of the bromide tracer was unachievable due to possible cryoturbation or other redistribution processes during freeze-up. I feel that the findings of this paper could merit publication; however there are some very major revisions that are required, including substantial rewriting. As it is written, the paper does adequately place this study in the context of previous research and the results are not clearly defined. The abstract and conclusion need to be re-worked to identify the scientific observations that will benefit the hydrology community.

It is my understanding that the authors are claiming that lateral transport across the frost table after infiltration is the most important finding of their study. The idea that frost table topography controls subsurface runoff has been well documented in the literature (Morison *et al*., 2016, Helbig *et al*., 2013; Quinton *et al*., 2000; Wright *et al*., 2009) and should be acknowledged as such, instead of as a novel finding. It was surprising that the authors briefly cited some very relevant studies for general water balance estimates (*i.e.* Helbig *et al*., 2013 for evapotranspiration; Liljedahl *et al*., 2016 for biogeochemical comparisons; Quinton *et al*., 2000 for hydraulic conductivity), but did not mention these studies in their discussion of subsurface routing in Arctic environments (and specifically ice-wedge polygons). By citing these papers the authors demonstrate that they are aware of these studies, but for some reason do not frame their research in the context of work that has already been completed. In the abstract, the sentence, "Estimates of horizontal hydraulic conductivity were within the range of previous estimates of vertical conductivity, highlighting the importance of horizontal flow in these systems" appears to be the most conclusive sentence in the abstract but does not convince the reader of a novel finding. The main finding in the conclusion is that, "horizontal flow is important". After reading this paper, I have not been convinced that horizontal flow is 'important', nor do I have an idea of how important it is on the total flux of subsurface runoff. I am also not convinced that this study, as-is, will provide a basis to improve hydrological models. In making these claims, the authors should: a) quantify horizontal hydraulic conductivity rates (this could be done directly in the field); and b) identify lateral flow routing mechanisms and attempt to quantify a landscape flux to demonstrate the relevance to this study. To do this, the results and discussion sections should be re-written to better position the paper's objectives and the authors should consider upscaling their findings to the subcatchment scale. The discussion section should be better framed with more reference to existing literature. As

currently written, most of the discussion lacks references, with the exception of occasional sentences having many references (*i.e.* page 20, line 8). The discussion section is a major weakness of the paper and could be written much better. Specific comments are listed below.

*We appreciate the constructive and thorough review. The comments provided were very helpful in improving our manuscript. In particular, this review provided feedback that helped us to improve the presentation of our work in the context of existing research. Thank you.*

*Response to general comments:*

*We did not intend to imply that the influence of the frost table on subsurface hydrology is a novel finding. We have added additional discussion including some of the papers suggested by the reviewer. After careful readings, these papers do not demonstrate, but only infer lateral transport controlled by frost table. We therefore disagree that the role of the frost table as a control on lateral flux is well established. In addition, some of the papers cited by the reviewer are not specific to this circumstance, as they are not specific to ice-wedge polygons (Quinton et al., 2000; Wright et al, 2009). Two of the papers that are studies involving ice-wedge polygons do not investigate high-centered polygons (Helbig et al., 2013; Morison et al., 2016) as our study does. None of the papers mentioned above show flow conditions across individual polygons. Our study is unique in that it demonstrates where water is flowing across entire, individual polygons. Furthermore, no other study has suggested a conceptual model of what flow and transport look like across entire low- and high-center polygons. Further discussion of each recommended paper is provided below. Modeling comments and other issues from the general comments are addressed in the specific comments section.*

**Morison et al., 2016**
*While this study was conducted in the subarctic rather than Arctic, we do agree that this paper should be mentioned in the discussion of subsurface routing. This paper does mention that soil storage is directly linked to the general frost table **position** and therefore the frost table is related to runoff thresholds. While this paper does mention the frost table, no measurements of the frost table **topography** were conducted as in our paper. Sentence added on page 18, line 15:*

*"This observation is consistent with observations of low-centered polygons by Helbig et al. (2013) and studies of other Arctic landforms underlain by permafrost (Morison et al., 2016; Wright et al., 2009)."*

**Helbig et al., 2013**
*We do agree that this paper should be mentioned in the discussion of subsurface routing. However, this paper infers subsurface lateral flux from water balances, rather than by direct measurement as in our study. While this paper does mention frost table topography in passing, it focuses mostly on the thickness of the active layer as a control on subsurface flow. As with lateral flux, this paper only infers the role of the frost table. In contrast, our paper shows the*

*relationship between flow and the frost table using an actual GPR survey of the frost table. Furthermore, Helbig et al. discusses spatially and temporal heterogeneity of flow across different low-centered polygons or troughs, but does not within individual polygons as does our paper. In short, our paper helps to confirm speculation made by Helbig et al.*
*Sentence added on page 18, line 15:*

*"This observation is consistent with observations of low-centered polygons by Helbig et al. (2013) and studies of other Arctic landforms underlain by permafrost (Morison et al., 2016; Wright et al., 2009)."*

*Quinton et al., 2000*
*While we do cite this paper for hydraulic conductivity, we disagree that it makes the point that frost table topography controls subsurface runoff. Overall, this paper emphasizes that lateral flow occurs in the unfrozen peat layer. Furthermore, the study sites in this paper are hummock-covered hillslopes which are likely to have different frost table topography than ice-wedge polygons.*

*Wright et al., 2009*
*Thanks for suggesting this reference. Although the landforms studied in this paper were peat plateaus and not ice-wedge polygons, we agree that this paper does document the control of the frost table on subsurface flow. We have acknowledged as much on page 18, line 15:*

*"This observation is consistent with observations of low-centered polygons by Helbig et al. (2013) and studies of other Arctic landforms underlain by permafrost (Morison et al., 2016; Wright et al., 2009)."*

*Liljedahl et al., 2016*
*We disagree that this paper merits mention for subsurface routing in Arctic environments. While this paper mentions hydrological states in relation to morphological state, it does not make any explicit statements about subsurface routing of water.*

**Specific comments:**

**Page 1 lines 29-32**: List references after each point instead of at end of the sentence. For example, "… as it affects hydrology (hydrology refs), biogeochemical transformations (biogeochemical refs)" etc.
Agree. Text modified – page 1, line 29 :

"Permafrost degradation is of primary concern in the Arctic, as it affects hydrology (Jorgenson et al., 2010; Liljedahl et al., 2011; Zona et al., 2011a), biogeochemical transformations (Heikoop et al., 2015; Lara et al., 2015; Newman et al., 2015;), and human infrastructure (Andersland et al., 2003; Hinzman et al., 2013)."

**Page 1 line 32:** How is 'northern' Arctic permafrost zone defined? All Arctic landscapes are northern, are you referring to the northernmost Arctic landscapes?

*Agreed. Text modified:*

*"The northernmost Arctic permafrost zone covers twenty-four percent of the landmass in the northern hemisphere and stores an estimated 1.7 billion tons of organic carbon"*

**Page 2, lines 19-20**: I struggle to understand the notation of the 'relative' roles of vertical and horizontal fluxes and that no other studies have been conducted toward quantifying this. It is generally accepted that in permafrost environments precipitation inputs: 1) infiltrate organic soils; 2) percolate to the frost table; and 3) produce lateral runoff where it is routed in accordance with frost table topography and is governed by fill-and-spill. There has been considerable work evaluating this principle, and other bodies of work that have evaluated subsurface runoff through ice wedge polygon terrain at the landscape scale (Helbig *et al.*, 2013; Liljedahl *et al.*, 2016).

*As discussed earlier, we disagree with the assertion that this is generally accepted. This has been speculated or implied by other papers, but not directly measured or demonstrated. One of the purposes of this study is to test whether this is the case. Unlike the studies cited by the reviewer, our study actually verifies lateral subsurface flow paths exist and quantifies horizontal flux. However, we think that it is reasonable to change "no studies" to "few studies." See previous response with discussion of these papers.*

*Text modified **(pg2, lines 22-23)**: "no studies" to "few studies" and clarified that we are referring to the differences between low- and high- centered polygons.*

I am also not convinced that if regional and pan-Arctic land models ignore horizontal fluxes that they would be well positioned to incorporate results of a study that document flow at the individual polygon scale. Furthermore, there are many hydrological models that include modules for subsurface routing, and even have options for different ways to parameterize that routing (*i.e.* Raven hydrological framework, Cold Regions Hydrological Model, Canadian Land Surface Scheme). I think the authors should also stress that this study seeks to better understand the differences in subsurface hydrology between low and high centered polygons, as this is a key component of the research.

*We agree with the reviewer. Models that do not incorporate horizontal fluxes are not well positioned to incorporate results of this study. Our point is that this may be problematic based on Helbig et al., 2013 and Liljedahl et al., 2016. Models that do incorporate lateral flow can benefit from the results of our study. Since polygons occupy such a large portion of the Arctic, their hydrologic behavior is incorporated via polygon subgrid heterogeneity in earth system models and experiments like ours are needed to support that effort. Unlike other studies, ours directly measures subsurface horizontal flux. The text has been modified for clarification – page 2, lines 18-21:*

*"Many studies have focused specifically on ice-wedge polygons, (Boike et al., 2008; Heikoop et al., 2015; Jorgenson et al., 2010; Lara et al., 2015; Newman et al., 2015) and provided much needed conceptualization (Helbig et al., 2013; Liljedahl et al., 2016). However, to our knowledge, few studies*

have been conducted toward quantifying the difference in relative roles of subsurface horizontal fluxes between low- and high-centered polygons…"

*We also agree with the assertion we should stress that this study seeks to better understand the differences in subsurface hydrology between low- and high-centered polygons.  While we have modified the text for clarification, it may be constructive to point out that there are numerous other places in the paper, particularly in the introduction, where we have stressed the difference in hydrology between polygon types as a key component of our research.  We have modified the text to clarify that we are referring to the differences between low- and high- centered polygons - **pg2, lines 22-23**:*

*"…studies, to our knowledge, have been conducted toward quantifying the difference in relative roles of subsurface vertical and horizontal fluxes between low- and high-centered polygons,…"*

**Page 3, line 10:** Again, routing mechanisms for lateral flow in polygonal terrain have been discussed. Helbig *et al*. (2013) conclude, "The prominent microtopography of the polygonal tundra strongly controls lateral flow and storage behavior".
*Helbig et al. did not compare low- and high-centered polygons.  The sentence referenced is in the context of the differences between low- and high- center polygons:*

*"The purpose of this paper is to examine how differently low- and high-centered polygons behave hydrologically, and evaluate the relative importance of vertical and horizontal flux within polygon systems (including the controls of the frost table and microtopography on subsurface hydrology)."*

**Page 4, line 5**: How representative are the properties of the polygons that were selected? Can you provide mean surface area and elevation (DEM?) for the study site?
*Agreed.  Text modified for clarity:  '…and have similar size and morphology.' was added*

*Page 4, line 8 also states:  …' the polygons selected are representative of a larger inventory of low- and high-center polygons being investigated by our team at this intensive study site…'*

*The very sentence in question refers to Figure 3 which is a DEM of both polygons, complete with topo lines, elevations, and a scale bar.*

**Page 5, line 2**: Are the pressure transducers absolute or vented? If the former, where is barometric pressure being collected?
*Text modified; added a sentence to this paragraph about barometric compensation:*

*"Barometric data was collected at the site and used to correct water level data for barometric effects."*

**Page 5, line 2:** How were the elevations of the well casings surveyed?
*Added a sentence to this paragraph about well survey:*

"All well casings were surveyed using a dGPS unit."

**Page 5, line 29:** How frequently was the sampler at the frost table moved down?
Text modified - added: "on a weekly basis" to the sentence

**Page 8, figure 5:** I would include this in the results section
*We appreciate the suggestion, but we want to place the figure where the 14 precipitation events are first mentioned in the methods. We do not feel the results section suffers from the absence of this figure or that it would be improved by its presence.*

**Page 9, line 34:** Why were values (and a subsequent range) for porosity used from the literature and not measured at the site?
*Sampling the polygons in the study for porosity would have potentially confounded the tracer test. Since a follow-on study is planned for the same polygons, porosity samples were not taken at the end of our experiment either.*

**Page 10, line 1:** There are numerous instances where this long list of references is used. It would be much more beneficial (and more informative) to include all of these studies in a table with their associated values for each parameter and then reference that table throughout the paper.
*Here we used the references to establish a range for porosity rather than presenting actual values from each reference, so we don't feel that a table is warranted. Over all, we appreciate this suggestion, but feel that this would result in an excessive number of figures/tables in the paper.*

**Page 10, line 2:** What was the time period over which the average head difference was calculated?
*Head difference was averaged over the course of observation during the first field season: 7/10/15 to 8/14/15. This was done because the velocities used to estimate hydraulic conductivity, based on first arrival of tracer, occurred in the first field season. To clarify, we modified the sentence in question:*

*"The change in hydraulic head was estimated by finding the average head difference, over the course of observation in 2015, between the well in the polygon center to the well nearest the sampler of interest."*

**Page 10, line 10:** How was the flux through organic soil calculated?
Vertical velocity was calculated, but actual flux was not.

**Page 10, line 19:** How does 'infiltration dominance' explain a rising water table? These sentences are worded awkwardly. This information would be much more clearly explained by showing a combined plot of cumulative evapotranspiration and cumulative precipitation.
*Infiltration dominance simply indicates that more water is infiltrating than evaporating, resulting in a rising water table.*

**Pages 10, 11, figures 6, 7:** These are very nice figures and display a lot of data in a format that is easy to read and digest.

*We appreciate the feedback.*

**Page 11, line 12:** I do not agree that the 2015 data shows that, "hydraulic gradients were often from the centre outward". I would argue that the hydraulic gradient was variable across the polygon. Also, this was not mentioned in the methods, but how were the wells surveyed and what was the error associated with these surveys? This may impact the hydraulic gradient measurements given that the elevation of all six water tables are within 40 cm.

*Agreed. Text modified – page 11, line 12: "For much of the 2015 thaw season, the water level in the center well was as high as or higher than three out of five of the trough wells, **indicating variable hydraulic gradients** across the polygon."*

*Text added to Materials & Methods section – page 5, line2: "All well casings were surveyed using a dGPS unit." The precision associated with our surveys ranged from 0.016 to 0.022 meters. We don't believe that this level of error would significantly change the interpretation of our data.*

**Page 11, line 17**: Given their close proximity, why would the purple and yellow trough wells on the HCP have water table differences of nearly one meter? Are there significant differences in soil type, topography, etc.?

*Yes, there is a significant difference in topography between these two wells – the yellow well is significantly lower.*

**Page 15, line 20:** Again, why not measure porosity of the mineral soil directly?

*Sampling the polygons in the study for porosity would have potentially confounded the tracer test. Since a follow-on study is planned for the same polygons, porosity samples were not taken at the end of our experiment either.*

**Page 15, line 21:** A range of 4.8 – 93.7% for possible tracer mass to leave the polygon is very high.

*The reason the range so large is because of a limited number of detections in the polygon. If one assumes the detections representative, which they are unlikely to be since tracer was not detected at most locations, then the estimate is high. We try to avoid being arbitrary by basing the estimate on the smallest and largest breakthrough curves.*

**Page 15, line 25:** Can you conclusively say that 'most' of the tracer remains in the centre of the LCP if your maximum estimate is that 93.7% left? Is there any way to improve this estimate? As-is, you cannot make this claim.

*To clarify, we are not actually saying that 93.7% of tracer left the polygon center. In the same paragraph, we clarify:*

*"This number is unrealistically high given that the breakthrough curve used in the estimate was incomplete and that the high bounding value used for mineral porosity was likely overestimated."*

*We also disagree with the assertion that we cannot make this claim. In the same paragraph, we explain that even using the smallest breakthrough curve to estimate mass likely results in an overestimate, thus justifying our claim:*

*"The smallest tracer mass estimated to have left the center, based on the smallest breakthrough curve, was 4.80%. This number can be considered a "maximum-minimum" and is likely an overestimate since tracer was not detected at all sampling locations around the polygon."*

**Page 16, lines 7-8:** Again, provide these references as a table with associated values
*As with the previous suggestion, we used the references to establish a range for porosity rather than using an actual number from each reference, so we don't feel that a table is warranted. Over all, we appreciate this suggestion, but feel that this would result in an excessive number of figures/tables in the paper.*

**Page 16, line 23:** Can you elaborate on the secondary porosity network and describe this more in Figure 12?
*Yes. The following four paragraphs elaborate on secondary porosity, preferential flow, and discuss Figure 12.*

**Page 16, line 33:** It may be worthwhile to include a discussion of heterogeneity and dual porosity in peat as well (I inferred that this section is restricted to the mineral soils).
*Agreed. The CT scans of cores include the top 40 cm of the soil profile, so they include peat. Text has been modified for clarity - page 16, line 35:*

*"These patterns reflect heterogeneity and dual porosity of the peat and mineral layers. "*

**Page 17, line 10:** Do the frost table elevations measured with a frost probe coincide with the GPR results?
*Yes, section 2.5 of Methods states: "strong relationship between the GPR signal travel time and Manual probe-based measurements of thaw layer thickness…" is mentioned in the GPR section of Materials and Methods.*

**Page 17, lines 8-18:** This is a good example of a paragraph that should be linked to existing literature that has evaluated the controls that the frost table exerts on subsurface runoff. A major weakness of this paper is that the discussion section does not integrate this study with other work to advance scientific understanding.
*Agreed. Text added; page 18, line 15:*

*"This observation is consistent with observations of low-centered polygons by Helbig et al. (2013) and studies of other Arctic landforms underlain by permafrost (Morison et al., 2016; Wright et al., 2009)."*

**Page 17, line 23:** "… as the frost table progressively deepens each year and these ice lenses thaw …" – This sentence implies that the active layer is becoming thicker every year. Is this the case? I have not seen a site where the active layer is thicker every year. Also, this section should contain mention of the ice-rich 'transient layer' described by Shur *et al* (2005). This discussion would be strengthened by including different values for hydraulic conductivity as the thawing front transitions from organic to mineral soil, and the controls that soil type has on subsurface runoff.

*We did not intend to imply that the active layer is thicker every year. Rather, we were referring to the seasonal thickening of the active layer.*

*Text modified for clarity - page 17, line 23:*

*"We speculate that, as the active layer progressively thickens each year and these ice lenses thaw, some of the resultant cracks remain open enough to create secondary porosity within the low-centered polygon."*

*Concerning the comment on hydraulic conductivity values relative to thaw front location in soil layers: In our study, we applied tracer to the organic layer and detected it at what we presume was the mineral layer and used tracer arrival times to estimate values of hydraulic conductivity. The nature of our experiment did not allow us to measure relative to the exact position of the boundary between organic and mineral layers without exploratory excavation that would have confounded the experiment.*

**Page 18, line 5:** What you are describing here is the transient layer (Shur *et al*, 2005). Again, a more detailed literature review is necessary to better frame the findings from this study.
*Agree. We added a sentence referencing the transient layer and Shur - Page 18, line 23:*

*"These structures are consistent with those found in the transient layer as described by Shur et al. (2005)."*

**Page 18, line 29:** Provide a reference for the statement that snowmelt only lasts between two and three weeks.
*Reference added:*

*'No samples were taken during snowmelt which typically no longer than two to three weeks (Hinzman et al., 1991)."*

**Page 18, section 4.2:** This appears to be a long-winded explanation of why part of the experiment failed, including explanations of various permafrost processes that have been explained before. This section could be greatly reduced and moved to the results section. Was there any monitoring of tracer concentration during freeze-up? This is a period of hydrologic activity that is often overlooked.
*We strongly disagree with the assertion that the experiment failed. Our tracer test has provided evidence that preferential flow is important. We think it is important discuss/hypothesize*

*possible drivers of this phenomena so future researchers (including ourselves) can think about what might be the controlling factor. It is necessary to discuss some of the possible mechanisms as these are things that will need to be addressed in future experiments.*

*While we appreciate the suggestion, we also do not agree that this section should be moved to the results section. This purpose of this section does not focus primarily on the results of our experiment, but is a discussion of possible drivers of observed phenomena.*

**Page 18, line 35:** The initial hypothesis that the interface of organic and mineral layers does not control horizontal flux may still be true. The authors should evaluate the relative roles of the horizontal flux while the frost table is in within the organic layer and when it descends to the mineral soil layer. The effect of subsurface runoff and the interplay between soil layers and frost table dynamics is a process that has been well documented, and should be referenced as such.

*We agree that there would be value in "evaluate(ing) the relative roles of the horizontal flux while the frost table is in within the organic layer and when it descends to the mineral soil layer." As noted page 18, line 37 the results of our study did not show any evidence of this. To do what is suggested is very difficult to do in an experiment such as ours without confounding the experiment since the organic layer is highly variable across the polygon. The intent of the shallow samplers (10 cm depth) was to get an idea of infiltration through the organic layer.*

**Page 19, figure 13:** In the high centered polygon, why is the vertical flux minimal/negligible? What happens to precipitation inputs if they do not infiltrate the soil column? If this is a conceptual diagram, should the water table in the centre of the polygon (LCP) not be higher than the trough if flow is directed outwards? Why is the major transport pathway to the right and not the left? There does not appear to be a difference in hydraulic gradient. Is this process limited by soil heterogeneity and differences in hydraulic conductivity? The rationale behind this diagram is not clearly evident.

*We thank the reviewer for addressing the wording of this caption. We have changed the wording in the caption to more accurately convey the meaning we intended:*

*"Dashed arrows indicate lower rates of flux than solid arrows. "*

*As for the low-centered polygon, we agree that water in the center of the polygon should be higher than in troughs and have updated figure 13.*

*That the major pathway is to the right and not the left is not intended to represent a specific position, but rather to indicate the spatial heterogeneity of flux. The flow arrows in the diagram are conceptual and not intended to represent a particular direction.*

**Page 19, lines 18-19:** Figure 6 does not indicate that water from polygon centers is distributed to troughs in LCPs. Actually, the data from 2016 indicates the opposite (as is stated in the results section). The discussion section should be written to better represent the data.

*To be clear, this sentence does not reference Figure 6 and specifies well responses to rain events:*

"Well responses indicated that water from polygon centers was redistributed to polygon troughs after rain events."

*Data depicted in Figure 6 implies that there is frequently a gradient from the center well to at least three trough wells, as the water level in the center well is often higher than these trough wells even in 2016. The comment seems to be based on the assumption that these wells are hydrologically connected at all times, but this may not be the case. If anything, Figure 6 infers that redistribution of rain water throughout the polygon is dependent on antecedent conditions. At any rate, Figure 6 is not intended for determining the hydrodynamic response of polygons to a rain event. Hence the well response and recovery analysis (section 2.7, 3.1, and table 1) included in our paper.*

**Page 19, line 25:** Would estimates of hydraulic conductivity not have been more reliable by completing pump/slug tests in the field?
*For the purposes of our experiment, pump tests would not have been reliable. With pump tests, it is not possible to separate vertical conductivity from horizontal conductivity. Also, pump tests would not reflect the variability in hydraulic conductivity across the polygon and would have been a substantial perturbation on the study sites.*

**Page 20, lines 5 and 6:** Can the impacts of freeze-up and thaw be elaborated? What effect does the two-sided freezing front have on subsurface hydrology in the thawed, saturated zone?
*This is discussed in the second paragraph of section 4.2:*

"Freeze-out is a process by which, as freeze-up progresses, most of the tracer remains in the aqueous component. In the Arctic, the active layer freezes from the top down and the bottom up simultaneously (although not necessarily at the same rate) (Cable, 2016). Thus, the tracer could have been redistributed within the soil profile as a result of freeze-out while remaining mobile in the unfrozen portion of the soil profile until freeze-up was complete. It has also been established that temperature gradients have the potential to cause redistribution of soil moisture (Hinzman et al., 1991; Painter, 2011; Schuh et al., 2017). During freeze-up, soil moisture in the active layer migrates toward freezing fronts (top and bottom) in a process known as cryosuction. The freeze-out and cryosuction processes could have a combined effect on redistribution of the tracer within the active layer of the polygons."

**Page 20, line 20:** I would not agree that field investigations are "almost totally lacking".
*Previous studies have shown/described some of this phenomenon, but we did a more direct interrogation. Perhaps "almost totally lacking" is an overstatement, but even one of the references provided by the reviewer agrees with our assessment. Helbig et al., 2013 (mentioned above) states that, "…despite their widespread occurrence in the Arctic, studies addressing specific hydrological processes of these landscapes related to their pronounced microtopography are still rare."*

**Page 20, line 22:** A major weakness of this study is that the lateral flux is not quantified. Indicating that lateral flow is 'important' is not a conclusion. A total flux (mm) from each polygon is needed if this work is to improve hydrological models.
*We think this is a gross overstatement. This study provides a conceptual model and tracer arrival times, both of which are extremely useful in improving hydrological models. Our study also demonstrates the existence preferential flow and that lateral velocities can be substantial, both of which can be helpful to models.*

**Page 20, line 24:** Is the Arctic Terrestrial Simulator the only hydrological model that these insights can help to improve? What is the rationale for including this model?
*The Arctic Terrestrial Simulator (ATS) performs calculations at the polygon scale and scales up to a watershed scale. The rational for including this model is that the authors of this paper are observational scientists who work in conjunction with a modeling team – we perform observational experiments that are used by the modeling team to improve models. While insights from this experiment can help improver other models, this experiment was done with the ATS in mind.*

*We have modified the text for clarity – page 3, line 15:*

"The Arctic Terrestrial Simulator performs calculations at the polygon scale and scales up to a watershed scale."

*Also, noted on page 3, line 13:*

"Insights from this study are intended to inform future work on the possible effects of permafrost degradation by improving the conceptualization used in the Arctic Terrestrial Simulator, developed by the Department of Energy at Los Alamos National Laboratory (Atchley et al., 2015; Painter et al., 2016) ."

**Page 20, line 27:** The final sentence is not a good concluding sentence for this paper.
*The last two sentences are now combined:*

"Additional work is also needed toward understanding controls on heterogeneity of flux in ice-wedge polygons, for example, the effect of ice lenses and cryoturbation on flux require further investigation."

**Technical corrections:**

**Page 1 lines 35-36:** The last two sentences are not sentences. Please rewrite.
Sentence now reads:

"Degree of soil saturation influences whether carbon is released as carbon dioxide or methane, thus highlighting the importance of understanding the hydrology of permafrost regions."

**Page 2 line 3:** "centers, rims, and **troughs**" Misspelled.
We have used the suggested edit.

**Page 10, line 16:** "From the beginning of July **until mid-August**…"

We have used the suggested edit.

**Page 13, line 20:** "Frost **table** depth"
We have used the suggested edit.

**Page 16, line 1:** "… tracer dynamic**s** …"
We have used the suggested edit.

**Page 16, line 21:** First sentence is not a sentence
We have used the suggested edit.

**Page 16, line 24:** "… range in horizontal **hydraulic** conductivity …"
We have used the suggested edit.

**Page 16, lines 34 -35:** Awkward sentence
Sentence now reads: "Patterns of vertical and horizontal density contrasts throughout the cores indicate the potential for preferential flow."

**Page 16, line 35:** "process", not processes
We have used the suggested edit.

**Page 16, line 36:** "… a potential cause **of** heterogeneity …"
We have used the suggested edit.

**Additional References:**

Morison, M.Q., M.L. Macrae, R.M. Petrone, and L. Fishback, L (2016), Seasonal dynamics in shallow freshwater pond-peatland hydrochemical interactions in a subarctic permafrost environment, *Hydrological Processes*, 31: 462-475, doi: 10.1002/hyp.11043.

Shur, Y., K.M. Hinkel, and F.E. Nelson (2005), The Transient Layer: Implications for Geocryology and Climate-Change Science, *Permafrost and Periglac. Process.*, 16, 5-17, doi:10.1002/ppp.518.

Wright, N., M. Hayashi, and W.L. Quinton (2009), Spatial and temporal variations in active layer thawing and their implication on runoff generation in peat-covered permafrost terrain, *Water Resources. Research.*, 45, W05414, doi:10.1029/2008WR006880

---

## Author Comment (AC2) · 26 Jul 2019

Please see supplemental .pdf file for response to referee comments

Please also note the supplement to this comment:
https://www.hydrol-earth-syst-sci-discuss.net/hess-2019-25/hess-2019-25-AC2-supplement.pdf

[Figure]

| | | | | | | | | Low-centered polygon frostline depths (cm) | | | | | | | | | | | |
|---|---|---|---|---|---|---|---|---|---|---|---|---|---|---|---|---|---|---|---|---|
| Date | N_rim | N_trough | NW_rim | NW_trough | E_rim | E_trough | SE_rim | SE_trough | S_rim | S_trough | SW_rim | SW_trough | W_rim | W_trough | NW_rim | NW_trough | AUX | C-1 | C-2 | C-3 |
| 7/11/2015 | 19.0 | 25.0 | 35.0 | 36.0 | 28.0 | 35.0 | 31.0 | 39.0 | 25.0 | 35.0 | 26.0 | 33.0 | 24.0 | 39.0 | 23.0 | 39.0 | n/a | 23.0 | 28.3 | 22.6 |
| 7/23/2015 | 24.0 | 29.0 | 41.0 | 36.0 | 35.0 | 35.0 | 35.0 | 46.5 | 32.0 | 35.0 | v.s. | 35.0 | 27.5 | 42.0 | 26.0 | 40.0 | 34.0 | 9.2 | 7.4 | 23.6 |
| 7/31/2015 | 26.5 | 29.0 | 42.0 | 36.0 | 35.5 | 39.5 | 37.0 | 47.0 | 33.0 | 35.0 | 35.5 | 36.0 | 31.0 | 42.0 | 28.5 | 41.5 | 35.0 | 11.3 | 8.5 | 24.6 |
| 8/7/2015 | 29.0 | 30.5 | 45.0 | 36.0 | 35.0 | 40.0 | 40.5 | 47.0 | 38.0 | 38.5 | 39.5 | 37.0 | 35.5 | 45.0 | 31.5 | 45.0 | 35.0 | 18.4 | 20.5 | 25.6 |
| 8/14/2015 | 29.5 | 31.5 | 45.0 | 37.5 | 36.5 | 44.0 | 44.5 | 47.0 | 44.5 | 39.0 | 42.0 | 41.0 | 36.5 | 46.5 | 35.0 | 46.0 | 38.0 | 20.2 | 25.1 | 26.6 |
| 9/12/2015 | 33.0 | 31.0 | 43.0 | 39.5 | 40.0 | 44.0 | 44.5 | 46.0 | 45.0 | 41.0 | 42.5 | 41.0 | 45.0 | 46.0 | 33.5 | 46.0 | 36.5 | 0.0 | 0.0 | 27.6 |
| 6/20/2016 | 9.0 | 10.0 | 20.0 | 16.0 | 10.0 | 13.5 | 13.5 | 17.5 | 9.0 | 12.0 | 9.0 | 13.0 | 9.5 | 12.5 | 10.0 | 14.5 | 10.5 | 0.0 | 0.0 | 28.6 |
| 6/23/2016 | 11.0 | 10.0 | 22.0 | 19.5 | 12.0 | 17.0 | 15.5 | 23.0 | 9.5 | 19.0 | 10.5 | 22.0 | 11.0 | 15.5 | 11.5 | 17.5 | 11.0 | 0.0 | 0.0 | 29.6 |
| 7/18/2016 | 27.5 | 32.0 | 38.0 | 39.5 | 33.0 | 35.0 | 37.0 | 45.0 | 29.0 | 33.0 | 34.5 | 34.5 | 26.5 | 34.0 | 25.0 | 32.0 | 31.3 | 0.0 | 0.0 | 30.6 |
| 9/2/2016 | 35.0 | 35.5 | 44.0 | 38.5 | 35.5 | 42.0 | 42.0 | 50.0 | 35.0 | 37.0 | 42.5 | 44.5 | 34.0 | 45.0 | 33.5 | 41.5 | 38.0 | 0.0 | 0.0 | 31.6 |

**Fig. 1.** Frost Table Depths LCP - to be added to Supplementary Information

| High-centered polygon frostline depths (cm) | | | | | | | | | | | | | | | | |
|---|---|---|---|---|---|---|---|---|---|---|---|---|---|---|---|---|
| Date | N_rim | N_trough | NE_rim | NE_trough | E_rim | E_trough | SE_rim | SE_trough | S_rim | S_trough | SW_rim | SW_trough | W_rim | W_trough | NW_rim | NW_trough | C-1 |
| 7/10/2015 | 21.5 | 21.0 | 22.0 | 25.0 | 25.0 | 25.0 | 27.0 | 20.0 | 24.0 | 21.0 | 24.0 | 25.0 | 23.0 | 21.0 | 25.0 | 25.0 | 17.7 |
| 7/23/2015 | N/A | 20.0 | N/A | 30.5 | 33.0 | 25.0 | N/A | 20.0 | N/A | 24.0 | N/A | 30.0 | N/A | 26.0 | N/A | 26.0 | N/A |
| 7/31/2015 | 25.5 | 25.0 | 30.0 | 33.0 | 35.5 | 28.5 | 38.5 | 23.5 | 34.0 | 25.0 | 41.0 | 33.5 | 27.0 | 29.5 | 27.5 | 33.0 | 24.7 |
| 8/7/2015 | 25.5 | 25.5 | 31.5 | 36.5 | 36.0 | 30.5 | 40.0 | 25.5 | 36.0 | 27.5 | 43.0 | 35.5 | 30.0 | 33.0 | 31.5 | 35.0 | 31.5 |
| 8/14/2015 | 25.5 | 26.0 | 35.0 | 36.5 | 36.5 | 32.0 | 41.0 | 25.5 | 36.0 | 31.0 | 43.5 | 35.5 | 31.0 | 33.0 | 32.0 | 36.0 | 31.8 |
| 9/12/2015 | 25.5 | 29.0 | 41.5 | 36.0 | 36.5 | 32.5 | 40.0 | 26.0 | 34.5 | 30.0 | 42.5 | 34.5 | 30.5 | 32.0 | 34.0 | 38.0 | 31.8 |
| 6/20/2016 | 6.5 | 12.0 | 10.5 | 11.0 | 13.0 | 13.0 | 10.5 | 12.0 | 10.5 | 8.5 | 14.0 | 19.0 | 11.5 | 7.0 | 8.0 | 14.0 | 11.7 |
| 6/23/2016 | 7.0 | 14.0 | 12.0 | 14.0 | 15.5 | 16.0 | 12.0 | 13.5 | 12.0 | 9.0 | 15.5 | 19.5 | 14.0 | 8.0 | 9.0 | 15.0 | 12.4 |
| 7/18/2016 | 18.0 | 26.5 | 29.5 | 28.0 | 39.0 | 26.0 | 29.0 | 21.5 | 32.0 | 22.0 | 36.0 | 35.0 | 33.0 | 26.5 | 24.5 | 28.0 | 23.0 |
| 9/2/2016 | 22.0 | 30.0 | 32.5 | 33.0 | 40.5 | 28.5 | 35.5 | 25.5 | 35.5 | 30.0 | 40.0 | 34.5 | 38.5 | 31.5 | 31.5 | 37.0 | 25.5 |

**Fig. 2.** Frost Table Depths HCP - to be added to Supplementary Information

**Supplement:**

**Response to Referee #2**

**General comments:**

The authors report on a bromide tracer experiment that took place in a single highcentered polygon and a single low-centered polygon in northern Alaska at the Barrow NGEE-Arctic site. The tracer was applied in 2015 and then measured through several sampling ports installed at different locations and depths across the polygon, including in adjacent troughs. The field conditions at the site are difficult and the thaw season is short; hence, the amount of data is sparse, as is the potential to conduct similar experiments across a larger number of polygons. The authors used a 1-D analytical solution to the convective-dispersion equation to estimate subsurface flow parameters, including vertical and lateral hydraulic conductivity (it appears that retardation factor was assumed based on a literature value). The comments below identify a number of areas that need further consideration. For example, the analytical solution assumes a point application, but the tracer in this case was applied to a large area; how should we interpret the boundary conditions used to determine lateral transport parameters? Also, the authors did not include any soil temperature in the manuscript, which would help identify freeze up and thaw, and the potential existence of ice lenses that would almost certainly impact the uniformity of vertical soil water flow. Without these data, the authors relied on conjecture to explain non-uniform transport behavior through the upper thawed soil. It is recommended that the authors include the time-series data on ice table depth, thus potentially helping here. Other comments are found below. Specific comments – comments called out by x/y, where x is page and y is line number

*We appreciate the time and careful consideration the reviewer has given to this manuscript. Comments regarding the analytical solution were particularly helpful in strengthening the position of our research.  All aspects of the general comments are addressed in the specific comments below.*

**Specific comments:**

3/8 – authors should clarify here that only one high-centered polygon and one lowcentered polygon were analyzed. As written, it appears that multiple polygons of both types were studied.
*Text has been modified:*

*"The purpose of this paper is to examine how differently a low- and high-centered polygon behave hydrologically…"*

4/15 – what was the total area into which bromide tracer was applied?
*Additional text added to include area of tracer application for each polygon:*

*"Blue circle indicates area of tracer application and encompasses the polygon center:  167.4 m² for the low-center polygon and 41.6 m² for the high-center polygon"*

5/8 – swap Figs. 4a and 4b to follow the order of call outs. Also, the description of the field setup using the silicon sheets doesn't appear on the subfigures. Suggest showing more detail in the schematic, so that the reader can note the silicon sheet, and that "surface" equals ground surface in current Fig. 4b.

*Agree – this is a good suggestion.  We have updated the figure and figure call outs according to feedback.*

5/26 – does the HCP have rims, as indicated in the sentence?
*Usual descriptions include centers, rims, troughs – we are trying to stick to established convention.*

6/18 – given that ponded water apparently existed in the LCP during tracer application, any information on soil water content to confirm that the thawed soil was fully wetted?
*Stating that the water was ponded during this time was an error – we have removed this language.*

6/30 – any soil temperature here or elsewhere at BEO that might be applicable here?  Also, it would be helpful for the authors to add a table (here or SI) that lists the frost table depth with time, especially given the importance to lateral transport and heterogeneity of the frost table depth.

*Yes, temperature has been collected in proximity to these polygons, but the data does not necessarily reflect frost table position.  However data from these other studies do show seasonal trends:*
https://ngee-arctic.ornl.gov/data/pages/NGA167.html
https://ngee-arctic.ornl.gov/data/pages/NGA118.html
*As for frost table data, we will place frost table depth with time in supplemental information.*

Figure 5 – suggest adding calendar date to either the x-axis or the caption, so that the reader can understand year-to-year variability of onset of thaw
*Caption text changed to include dates:*

*"Precipitation events, from July 3, 2015 to September 30, 2016, used in the calculation of characteristics of well response."*

8/10 – van Genuchten and Alves (1982) solution assumes 1D transport, or in the context of this experiment, a point application of tracer. How does the broad area of application square with this assumption? Was it only used to estimate velocities during that segment of the flowpath, and then a second calculation for estimating horizontal flow? How is lateral distance determined for those sampling clusters outside of the

application area? Also note that the van Genuchten and Alves reference on 24/33 is incomplete.

The reviewer is correct, the analytical solution by van Genuchten and Alves (1982) is for an infinite one-dimensional system, not for a broad area. In this manuscript, the analytical solution is used to describe the fate and transport of the tracer in a typical flow path with the boundary conditions imposed at the surface of the experimental domain. This conceptualization is a parsimonious approach to explore the first-order factors controlling fate and transport and time scales within this complex system. A more detailed modeling approach is out of the scope of this work, but it will be a future contribution. Finally, the lateral distances are estimated as the distance from the sampling cluster to the edge of the polygon center (tracer application area), a simplification used to estimate the order-of-magnitude of the solute arrival times and hydraulic conductivities.  Essentially, a point to point solution is appropriate in this case.

We have added the following language to the paper to emphasize this point and the assumptions of the analysis - Page 8, line 6:

"To this end, velocities were estimated by assuming that the transport of the tracer within the polygons can be approximated as a one-dimensional advective-dispersive problem with adsorption effects – a reasonable assumption given the lack of information and uncertainty in the spatial distribution of hydraulic parameters. This is a parsimonious approach to explore the first-order factors controlling fate and transport and time scales within this complex system."

We have also modified text on Page 9, line 2:
"…x [cm] is the lateral distance from the sampling nests to the edge of the tracer application area."

**Also note that the van Genuchten and Alves reference on 24/33 is incomplete**
*This is a good catch.  We have updated the reference.*

9/5 – check table 2. As presented, neither background concentrations nor tracer injection data are included
*The reviewer is correct.  We have removed the reference to Table 2 – this is a vestige from an earlier version of the paper.  Background concentrations and tracer injection data are presented in the following sentence (page 9, line 5).*

9/9 – the retardation factor for Korom's experiment were for sediment with a pH of between 5.1 and 5.7. According to Goldberg and Kabengi (2010, doi:10.2136/vzj2010.0028), retardation of bromide is very pH dependent. In some cases, bromide transport in soil with can lead to retardation factors significantly less than one (see for example Hills et al., 1991, WRR, paper 91WR015). How do the soil conditions at the Barrow site compare with those from Korom? Are the data robust enough to estimate R either through parameter estimation or other means? Given how R scales the tracer velocity, so more thought on this issue is warranted.
*We thank the reviewer for pointing this out.  The soil conditions at the Barrow site are comparable to those from Korom .  We have included the following for clarification:*

"With an average pH of 5.6 in the study area (Newman et al.), the retardation factor was approximated as R=1.56 (Korom, 2000) – a reasonable value given the pH in Korom's experiment was between 5.1 and 5.7."

9/23 – any particular reason why sampling and analyses occurred for only two years, when it became clear that tracer recovery would be so low?
*Yes, funding was limited. The extensive sampling array and analytical costs became prohibitive. Also, this paper is based on my Master's thesis and my Master's degree program came to an end.*

9/25 – here and elsewhere, it is suggested that the authors refer to tracer application in the polygon interior, rather than application in the polygon center. Indeed, most of the interior of the polygon received tracer, rather than a point application.
*It is a well-established convention to refer to this microtopographic feature of the ice-wedge polygon as the polygon center. We wish to adhere to this convention in order to avoid confusion.*

10/8 – if I understand the narrative correctly, the polygon was represented as an idealized vertical cylinder, and the flux was estimated through the bottom of the cylinder based on measurements from the rhizon nests, is that correct? Was the flux then used as initial conditions for the lateral flow the nests outside of the cylinder?
*To clarify, the polygons were represented as idealized cylinders and lateral flux was estimated through the sides of the cylinder, not the bottom. This flux estimate was based on measurements from the most distal rhizon nests (troughs). In other words, the most distal rhizon nests would be located at the sides of the cylinder (edge of the polygon) rather than outside of the cylinder, so flux was not used as an initial condition for flow to the very same rhizon nests. We have modified the text for clarification:*

*Page 10, line 10:*
*"Second, flux was calculated through the side of the cylinder...."*

11 (general) – the authors seem to bounce from LCP and HCP results, first referring to water levels, then to delta H values for both. It would be easier to discuss LCP first, then HCP second
*Agree – We have rearranged this section as the reviewer suggests.*

Figure 8 – Fig. 8a shows location of GPR measurements and results, but not frost table slope, and Fig. 8b shows frost table slope but not GPR measurements. Could both results be shown for both polygons?
*Both Figure 8a and 8b show GPR measurements. As explained in section 2.5, the spatial density of the GPR data at the high-center polygon was sufficient enough to produce an elevation map of the frost table, which is what Figure 8b depicts (see legend). As for Figure 8a, the spatial*

*density of the GPR survey was not sufficient to produce an elevation map of the frost table. Therefore, a frost table slope could not be determined.*

*Text modified in figure caption to clarify – page 12, line 14:*

*"Note that transect lines indicate frost table elevation at the low-centerd polygon (a) while topo lines indicate frost table elevation at the high-centered polygon (b)."*

14/2 – replace "Surface" with "Trough"
*This sentence is referring to surface water in the troughs and it is important to distinguish this from subsurface water collected in the troughs.  We have modified this sentence for clarity:*

*"Surface water samples collected from troughs during 2016 did not show a clear trend of increasing tracer concentration (Fig 10a)."*

14/5 – similar to the comment above, any soil temperature data that could help interpret these results in successive years? The reduced concentration from the end of 2015 to the beginning of 2016 is puzzling and potentially indicates transport even though water appeared frozen.
*We appreciate the suggestion.  As stated above, we will place frost table depth with time in supplemental information.  As discussed in section 4.2 of the paper, there does appear to be a reduced concentration from the end of 2015 to the beginning of 2016 and we have provided some possible reasons why this occurred.*

15/22 – are the authors stating that tracer recovery of 4.80% is actually a high estimate?
*Yes, we are stating that we consider 4.80% to be a high estimate for the lower bounding value. Because of spatial variability and different tracer arrival times and concentrations, the mass balance is presented as a range with 4.80% at the low end of this range.*

15/25 – when authors refer to polygon 'center,' is this really the polygon 'interior?'
*We thank the reviewer for this comment.  We have modified the text for clarification:*

*"Even though these estimates have large uncertainties, it appears that most of the tracer remains within the interior of both polygon centers."*

16/21 – authors are using either preferential flowpaths or heterogeneity of subsurface media as possible reasons for non-uniform vertical flow, or bypass flow around shallow samplers. A third explanation here is that the soil has undergone partial melting or partial freezing, reducing liquid water-filled transport pathways, and facilitating transport through specific pathways. This might also explain why tracers are changing concentration so drastically between thaw seasons.
*We have added this to our discussion of ice lenses and CT scans of frozen cores:*

*"These patterns may also be indicative of partial melting or partial freezing of the soil profile as a driver of heterogeneous flow."*

Figure 12 – though the figures are interesting, there's not enough explanation behind them to know whether the conditions represented by these images are the same as those observed at the traced polygons. It is suggested that the authors either more closely tie the images from Romanovsky to the site being reported on here, or consider removing the figures altogether.

*These cores were collected within same general study area (BEO) in similar polygons. We obviously could not core before our experiment because it would have confounded the tracer test.*

---

## Referee Report (RR1)

Understanding the Relative Importance of Vertical and Horizontal Flow in Ice-Wedge Polygons

Wales *et al*. (2019)

**General Comments:**
The authors have made substantive changes to the manuscript as suggested by reviewers. All major concerns have been addressed either in-text or in response to reviewer comments. I commend the authors on the additional work put into this text. This study summarizes field work from a logistically challenging region where field data and intensive studies are limited. I believe that this work will be a valuable contribution to the scientific literature.

**Minor Comments:**
The authors may consider including the raw GPR radargram images as supplementary material.

**Page 4, lines 14-16**: Include a line describing the potential for ice lenses (as this is an important piece in the discussion)

**Page 13, line 11**: Figure 8 caption, change 'topo' to topographic.

**Page 17, lines 7-8**: A table here would be a valuable resource to the reader, depicting the: a) study (authors); b) study site (with latitude); and c) range of hydraulic conductivity

**Page 19, line 10**: This point here may be more appropriate to include the reference of Shur *et al*. (2005) discussing the transient layer, as this ice-rich layer forms at the interface of the permafrost table and the base of the active layer.

**Page 19, line 29**: Consider rephrasing to: "temperature **and vapour** gradients"

**Page 19, line 31**: "… in a process known as cryosuction…" consider rephrasing to: "During freeze-up, cryosuction allows for soil moisture to migrate towards the freezing front"

**Page 19, line 35**: "which typically no longer than", change to: "which typically **lasts** no longer than"

---

## Author Response (AR2)

**Response to Referee #1**

Understanding the Relative Importance of Vertical and Horizontal Flow in Ice-Wedge Polygons
Wales *et al*. (2019)

**General Comments:**
The authors have made substantive changes to the manuscript as suggested by reviewers. All major concerns have been addressed either in-text or in response to reviewer comments. I commend the authors on the additional work put into this text. This study summarizes field work from a logistically challenging region where field data and intensive studies are limited. I believe that this work will be a valuable contribution to the scientific literature.
We are grateful for the careful review.  The comments/edits provided were useful in improving our manuscript.

Response to comments:
**Minor Comments:**
The authors may consider including the raw GPR radargram images as supplementary material.
We appreciate the suggestion. The raw and processed GPR data will be included in a public data DOI record that is in progress and will be made available through the NGEE Arctic website.

**Page 4, lines 14-16**: Include a line describing the potential for ice lenses (as this is an important piece in the discussion)
Text modified – page 4, line 16:
"Ice lenses can form during freeze-up, especially in the mineral soil."

**Page 13, line 11**: Figure 8 caption, change 'topo' to topographic.
Text modified – page 13, line 11:
Changed "topo" to "topographic"

**Page 17, lines 7-8**: A table here would be a valuable resource to the reader, depicting the: a) study (authors); b) study site (with latitude); and c) range of hydraulic conductivity
We appreciate the suggestion.  We have added a table with author, $K_{sat}$, and porosity values to supplementary material.

**Page 19, line 10**: This point here may be more appropriate to include the reference of Shur *et al*. (2005) discussing the transient layer, as this ice-rich layer forms at the interface of the permafrost table and the base of the active layer.
Agree – reference added

**Page 19, line 29**: Consider rephrasing to: "temperature **and vapour** gradients"
Text modified as suggested.

**Page 19, line 31**: "… in a process known as cryosuction…" consider rephrasing to: "During freeze-up, cryosuction allows for soil moisture to migrate towards the freezing front"

Text modified as suggested.

**Page 19, line 35**: "which typically no longer than", change to: "which typically **lasts** no longer than"
Text modified as suggested.

**Response to Referee #2**

This is a second review for this manuscript. In general, the authors have done an acceptable job addressing earlier comments and questions, both from me and the other reviewer. The review is primarily focused on responses to a few comments raised earlier.
We thank the reviewer for the thoughtfulness put toward this manuscript. The specific comments were helpful in refining the final draft.

Response to comments:
Specific comments – comments called out by x/y, where x is page and y is line number in the revised manuscript.

7/1, 10/12 and elsewhere – I disagree with the authors regarding the use of the term 'center' versus 'interior' to identify tracer application, which was a point of confusion to this reviewer in the original manuscript. I don't see the advantage in following conventional terms to describe an unconventional experiment. Indeed, had the tracer been applied to the center of the polygon, as stated in the paper, then the assumptions underlying van Genuchten and Alves solutions would be have been valid and easier to justify, rather than using the edge of the vertical cylinder that represented the edge of the application area. Using the term 'center' is inaccurate, regardless of whether it follows convention.
Text modified – 'center' changed to 'interior' when in context of tracer application.

7/7 – That the ponded water didn't exist when tracer was applied certainly simplifies the analysis, as least from the standpoint of dilution, but the basic question remains: is water content data available? If not, how was liquid-filled porosity handled in the velocity term for van Genuchten and Alves? What is the impact of partially saturated material on the vertical migration of tracer down to the ice table? How are you estimating pore water velocity?
Estimation of pore water velocity is explained on page 10, line 5. Text was modified to clarify use of velocity term for van Genuchten and Alves – page 10, line 5: added "…and assumes full saturation."

There are no measured soil water content data available from the polygons, but saturation is a reasonable assumption. Multiple field observations were that the LCP was saturated at all times, even when there was no surface water ponding. As for the HCP, observations where that the mineral layer was always saturated and that the peat layer in the polygon center was frequently saturated. Squishy, wet soil conditions at the field site were nearly always present. Water level data from observation wells corroborates these observations. The limited volume

and short duration of unsaturated conditions in the HCP increases uncertainty in the velocity estimation, but the effect is considered minor, especially since the velocities are considered estimates.

16/21 – regarding the issue of experiment length and tracer recovery, it is indeed an unfortunate reality that funding is not infinite and neither is the time available to complete a MS thesis. That said, these two factors did impact the results of the experiments, and our understanding of how heterogeneity might impact shallow subsurface transport, by limiting the number of polygons that could be tested and by not running the experiment long enough to recover enough tracer to avoid using ~5% recover as a high estimate on the lower range. What does this mean going forward? How long should the experiment have lasted? What value could be expected by conducting a sufficiently long experiment?
Future tracer studies of this kind should consider that an extended multi-year sampling approach will likely be necessary to adequately capture complete breakthrough curves.  Such experiments would highlight the importance of interannual variations and improve understanding of the impacts of multiple winter freeze up events on transport.

Text modified – page 21, lines 32-38

[revised manuscript text omitted]

---

## Author Response (AR3)

**Response to Editor's Edits**

**Specific comments:**

**Page 1 line 4**: One of the coauthors was inadvertently left off between versions and we have added him back in.
Agree. Text modified – page 1, line 4: "Florian Soom[4]" added

**Done - Page 1 line 15:**
"hydrology" changed to "hydrological"

**Done - Page 1 line 15:**
"hydrology" changed to "hydrological"

**Done - Page 1 line 21:**
"of fluxes" deleted

**Done – Figure 1:**
Added reference to supplementary material

**Done - Page 7 line 2:**
Added application area for each polygon:
"167.4 $m^2$ for the low-center polygon and 41.6$m^2$ in the high-center polygon"

**Done - Page 7 line 10:**
Added text: "…in the MacroRhizon samplers"

**Done – Figure 5:**
Changed axis label to mm/day

**Done - Page 9 line 14:**
Changed "Equations" to "Eqs."

**Done – Figure 6:**
Axis label changed from daily to hourly precip – was a typo. This was not an average. Precip data was collected from only one source (see section 2.2 Observational network).

**Done – Figure 7:**
Axis label changed from daily to hourly precip – was a typo. This was not an average. Precip data was collected from only one source (see section 2.2 Observational network).

**Done – Figure 8**

Map and Legend have been adjusted.  The lower bound elevation value in the previous version was from a larger map of the area.  The values now make sense when using values only from the area in the figure. The lower bound for ground surface is now 4.5.

Also added reference to supplementary material in Figure 8 caption.

**Done - Page 19 line 31:**
Changed "point" to "front"
We use the term 'freezing front' rather than 'frost table' because bidirectional freezing occurs in this region and the 'freezing front' is not necessarily only at the frost table.